# The gut hormone Allatostatin C/Somatostatin regulates food intake and metabolic homeostasis under nutrient stress

Olga Kubrak[1,2], Takashi Koyama [1,2], Nadja Ahrentløv [1,2], Line Jensen[1], Alina Malita[1], Muhammad T. Naseem [1], Mette Lassen[1], Stanislav Nagy[1], Michael J. Texada [1], Kenneth V. Halberg [1] & Kim Rewitz [1✉]

The intestine is a central regulator of metabolic homeostasis. Dietary inputs are absorbed through the gut, which senses their nutritional value and relays hormonal information to other organs to coordinate systemic energy balance. However, the gut-derived hormones affecting metabolic and behavioral responses are poorly defined. Here we show that the endocrine cells of the *Drosophila* gut sense nutrient stress through a mechanism that involves the TOR pathway and in response secrete the peptide hormone allatostatin C, a *Drosophila* somatostatin homolog. Gut-derived allatostatin C induces secretion of glucagon-like adipokinetic hormone to coordinate food intake and energy mobilization. Loss of gut *Allatostatin C* or its receptor in the adipokinetic-hormone-producing cells impairs lipid and sugar mobilization during fasting, leading to hypoglycemia. Our findings illustrate a nutrient-responsive endocrine mechanism that maintains energy homeostasis under nutrient-stress conditions, a function that is essential to health and whose failure can lead to metabolic disorders.

[1] Department of Biology, University of Copenhagen, 2100 Copenhagen O, Denmark. [2]These authors contributed equally: Olga Kubrak, Takashi Koyama, Nadja Ahrentløv. ✉email: Kim.Rewitz@bio.ku.dk

Organisms' survival depends on their ability to maintain energetic homeostasis[1], and deficiency in this ability is implicated in the pathogenesis of human diseases such as diabetes and obesity. Maintaining whole-body energy balance requires coordination of nutrient sensing, energy metabolism, and nutrient intake, functions distributed across discrete organs. Such coordination is achieved through the exchange of metabolic information between organs, mediated by nutrient-responsive hormones. These hormones are secreted by specialized tissues that sense changes in internal and external nutritional conditions, and underlie inter-tissue communication that maintains systemic energy homeostasis via the regulation of appetite and metabolism. Defects in this exchange of nutritional information between tissues can lead to metabolic dysfunction. Thus, characterizing the network of hormones that convey nutrient availability and govern energy homeostasis through the control of food intake and energy expenditure is critical for understanding human metabolic disorders and for developing future therapeutics.

The gut is the largest endocrine organ and a central coordinator of systemic energy homeostasis[2]. Ingested nutrients are absorbed through this organ, which also senses nutritional value and relays metabolic information to other tissues via the secretion of hormones that are released into circulation from enteroendocrine cells (EECs) and coordinate body-wide processes[3]. The EECs, therefore, play an important role in balancing energy homeostasis through crosstalk with regulatory networks that regulate appetite and metabolism. In the mammalian intestine, for example, the EECs release ghrelin during fasting, which signals directly to the brain to regulate appetite and energy homeostasis[4]. Glucagon-like peptide 1 (GLP-1) is secreted by the EECs in a glucose-dependent manner and plays a key role in regulating glucose metabolism[5]. Together GLP-1 and gastric inhibitory peptide (GIP) underlie the so-called incretin effect, through which oral glucose consumption induces a much stronger insulin-secretory response than intravenous administration of glucose. GLP-1 is secreted from the intestine in response to ingestion of glucose and acts directly on the pancreatic β-cells to potentiate glucose-dependent insulin secretion. This incretin effect is diminished in type-2 diabetes patients, and the use of incretin-hormone therapy is of great medical importance in the treatment of metabolic disorders.

The *Drosophila* gut is emerging as an important model for studies of the systemic effects of intestinal homeostasis. The fly gut shares structural and functional features with the mammalian intestine, and many of the systems that mediate nutrient signaling and control metabolism are evolutionarily ancient and conserved[1,6,7]. In *Drosophila*, metabolic homeostasis is regulated by hormones similar to insulin and glucagon. In mammals, insulin promotes the cellular uptake and conversion of excess nutrients into glycogen and lipids, which are stored in liver and adipose tissues under nutrient-replete conditions. Under conditions of nutrient scarcity, mammalian glucagon induces the mobilization of stored energy to provide the means to sustain glycemic control and basic physiological body functions. Likewise, flies express several conserved insulin-like peptides (DILPs) in insulin-producing cells (IPCs) of the brain, which are believed to be functionally and evolutionarily related to mammalian pancreatic β cells[8,9]. During nutrient abundance, these DILPs promote growth and metabolism through their activation of a single insulin receptor. Adipokinetic hormone (AKH) is functionally related to mammalian glucagon and is known to act antagonistically to insulin. Under conditions of nutritional scarcity, AKH is secreted from AKH-producing cells (APCs) of the corpora cardiaca (CC), neurosecretory tissues analogous with pancreatic α-cells, to promote energy mobilization and food-seeking behaviors. The *Drosophila* fat body is the main energy-storage organ and is functionally analogous to the mammalian liver and adipose tissues. During nutrient-stress conditions, AKH acts on the fat body via its receptor AkhR to promote the breakdown of stored lipids and carbohydrates to prevent hypoglycemia[10]. AKH also acts on a small set of arousal-promoting neurons in the brain to promote food-seeking behavior[11].

Like the EECs of the mammalian intestine, those of the *Drosophila* gut secrete a cocktail of endocrine factors into the hemolymph (insect circulatory fluid) that relay information to other target organs. Although evidence suggests that both mammalian and *Drosophila* guts produce a large number of hormones, the functions of these signals, as well as the information they convey, remain poorly defined. In the adult fly gut, EECs are marked by expression of the transcription factor Prospero (Pros) and produce more than a dozen known peptide hormones[12–15]. These include Allatostatin A (AstA), AstC, Bursicon alpha (Burs), neuropeptide F (NPF), short neuropeptide F (sNPF), diuretic hormone 31 (Dh31), pigment-dispersing factor (PDF), CCHamides 1 and 2, Myoinhibiting peptide precursor (MIP or AstB), Orcokinin B, and Tachykinin (TK). However, the function of these gut hormones is known in only a few cases. Gut-derived TK regulates intestinal lipid homeostasis[16], and NPF from the gut governs mating-induced germline stem-cell proliferation and metabolism[17,18]. Burs is released from EECs in response to nutrient intake and regulates energy homeostasis through a neuronal relay that modulates AKH signaling[19]. Dh31 upregulates gut motility in response to enteropathic infection[20]. Although incretin hormones and insulin secretion have been extensively studied in both mammals and *Drosophila*, the regulatory mechanisms controlling the secretion of glucagon remain poorly defined, and nutrient-responsive gut hormones with the ability to promote glucagon secretion are not known. To characterize factors secreted by the intestine that are important for energy homeostasis, we performed a genetic screen for gut hormones that affect metabolic stress responses.

We show here that intestinal nutrient signaling is mediated by AstC and AstC receptor 2 (AstC-R2), which are homologous with mammalian somatostatin and its receptors[21]. AstC is released from EECs via a mechanism involving Target of Rapamycin (TOR) in response to nutrient deprivation and is required for the mobilization of stored lipids and carbohydrates to prevent hypoglycemia during periods of starvation. Gut-derived AstC signals the fasting state to the APCs via AstC-R2 and promotes food intake and energy mobilization by stimulating AKH secretion. These findings demonstrate that in response to fasting, AstC is released by EECs to coordinate systemic metabolic-stress responses, regulating energy homeostasis through the inter-organ regulation of AKH secretion. Our findings illustrate a mechanism by which gut-mediated nutrient sensing is required for the maintenance of organismal energy homeostasis during nutritionally challenging conditions. In response to fasting, gut-derived AstC enhances hypoglycemia-induced AKH secretion – conceptually similar to the role of incretin hormones in promoting insulin secretion when glucose is consumed – and AstC is key to maintaining metabolic homeostasis during nutrient-deprived states. Dysregulation of such gut-mediated signaling could contribute to metabolic disorders.

## Results
**Gut AstC signaling regulates organismal resistance to nutritional challenges**. To identify intestinal hormones involved in nutrient sensing and in maintaining energy homeostasis during nutrient stress, we used RNAi to knock down secreted factors in the EECs of the adult *Drosophila* midgut and tested for changes in organismal starvation resistance (Fig. 1a). We restricted the RNAi

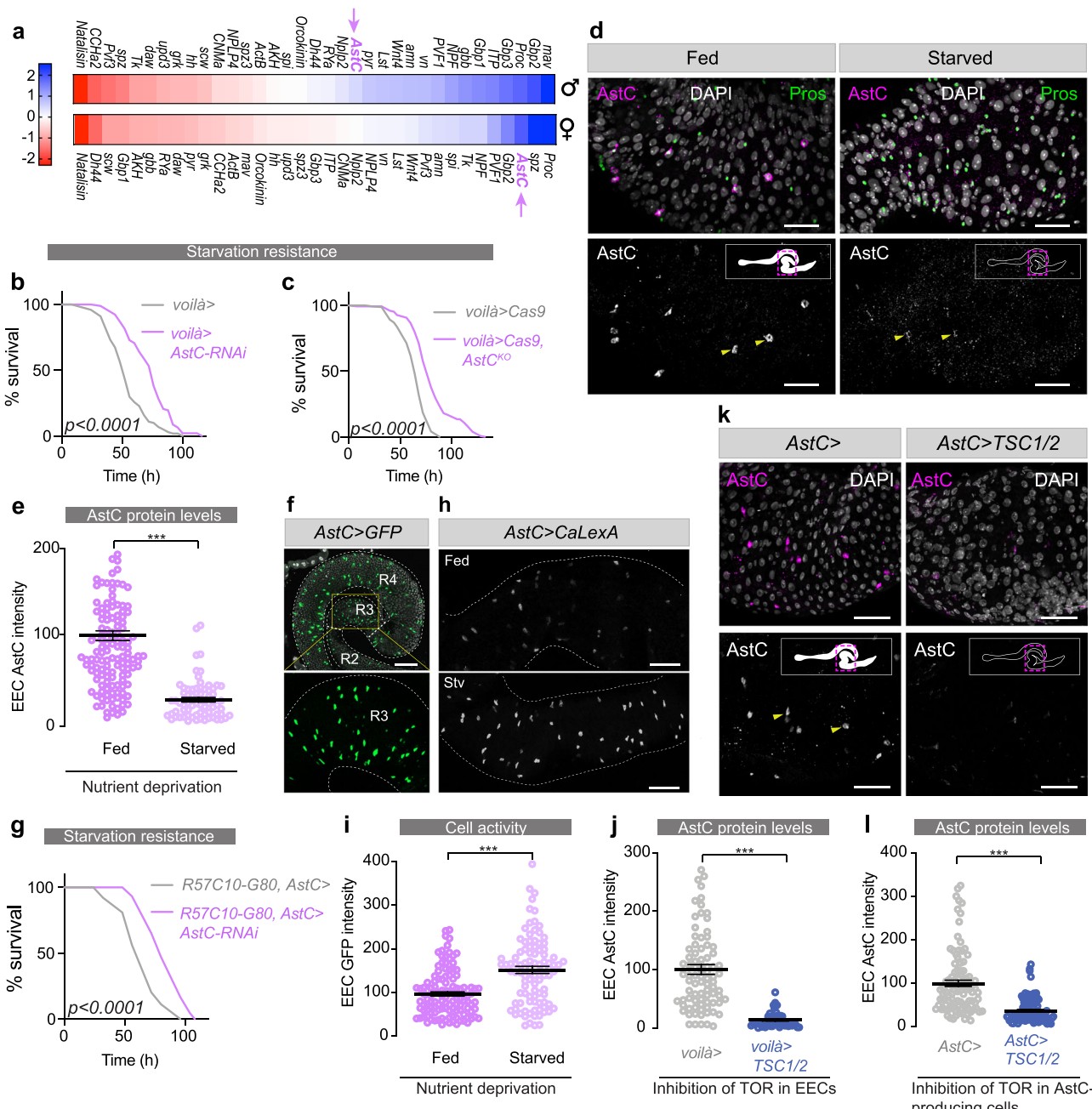

**Fig. 1 AstC secretion from the gut is nutrient-responsive and regulates organismal resistance to nutrient stress. a** A screen of adult-midgut peptides identified those that affect starvation survival. Color scale indicates the number of standard deviations from the mean of all lines. Adult-restricted EEC-specific RNAi (**b**) or CRISPR-mediated knockout (**c**) of *AstC* prolongs female starvation survival ($p < 0.0001$). **d** Starvation induces AstC release from EECs (marked with anti-Pros). AstC staining is prominent in the R3 region (middle midgut) in fed females and is strongly reduced under starvation. *AstC* expression remains constant, Supplementary Fig. S1d. Representative of nine guts. Scale bars 50 μm. **e** AstC staining in R3 drops with starvation ($p < 0.0001$; results represent 9 guts, $n = 145$ fed and 72 starved cells). **f** AstC-expressing cells are abundant in female R3 and R4 (*AstC > GFP*, green) and less common in R2. Bottom: an enlargement of R3, in which calcium-LexA was measured. Image is representative of six guts. Scale bar 100 μm. **g** Female starvation survival is prolonged ($p < 0.0001$) with *AstC* knockdown in the AstC-producing EECs of the gut. **h** Starvation activates AstC-expressing EECs of R3, as measured by calcium-induced GFP expression. Scale bars 50 μm. These data represent analysis of 6 guts, $n = 120$ fed and 100 starved cells; $p < 0.0001$, and are quantified in (**i**). **j** TOR activity inhibits EEC AstC release. Expression of *TSC1/2* in EECs for 6 h leads to a strong reduction in R3 AstC staining (results represent analysis of guts from at least 8 animals, $n = 85$ *voilà>* and *voilà>TSC1/2* cells; $p < 0.0001$), despite an increase in *AstC* transcript levels (Supplementary Fig. S1h). **k** Inhibition of TOR in the AstC-producing cells alone using *AstC>* to conditionally express *TSC1/2* also strongly reduced R3 EEC AstC staining, quantified in (**l**) (at least 9 guts, $n = 99$ *AstC >* and 99 *AstC>TSC1/2* cells; $p < 0.0001$). Scale bars 50 μm. Error bars indicate SEM. ***$p < 0.001$. Statistical significance was determined using Kaplan–Meier log-rank tests (**b**, **c**, **g**) or two-tailed Mann–Whitney $U$ test (**e**, **i**, **j**, **l**). Source data are provided as a Source data file.

effect to the adult stage using a GAL4 driver (*voilà-GAL4*) that targets the EECs in combination with *Tubulin*-driven temperature-sensitive (TS) GAL80 (GAL80$^{TS}$) expression, together referred to as *voilà>*. We found that EEC knockdown of some factors including *Natalisin* resulted in sensitivity to starvation, while depleting others such as *Proctolin* and *AstC* enhanced starvation resistance. We focused our attention on AstC, since *UAS-AstC-RNAi* (*AstC-RNAi*)-mediated knockdown of this factor in the adult EECs (*voilà>AstC-RNAi*) caused a striking prolongation of starvation survival in females, but not in males, compared to *voilà>* controls (Fig. 1b and Supplementary Fig. 1a). We also excluded contributing effects from the RNAi line alone (Supplementary Fig. 1b). Sex-specific differences in metabolism are apparent in both humans and flies, but the underlying mechanisms are generally not well understood. In *Drosophila*, male-specific fat metabolism is related to dimorphic expression of the lipase Brummer, and male-biased intestinal signaling has been related to feeding and sperm maturation[22,23], but the mechanisms underlying female-specific metabolic control remain to be characterized. To further confirm this phenotype, we used tissue-specific somatic CRISPR/Cas9-mediated gene editing to disrupt the *AstC* locus in the EECs of adult flies. For this purpose, we generated transgenic animals with UAS-inducible expression of a pair of guide RNAs (gRNAs) targeting *AstC*, which was driven with *voilà>UAS-Cas9* (*voilà>Cas9*) containing GAL80$^{TS}$ to restrict effects to the adult stage. Deletion of *AstC* in the EECs in the adult stage (*voilà>Cas9, AstC$^{KO}$*) led to a strong increase in resistance to starvation in females compared to controls (*voilà>Cas9*), but not in males (Fig. 1c and Supplementary Fig. 1c), thus confirming the effect observed for RNAi-mediated knockdown. This phenotype suggests a post-developmental role of gut AstC in organismal responses to nutrient deprivation.

**Nutrient sensing in the EECs promotes release of AstC upon nutritional restriction.** To be able to regulate organismal metabolism and food intake according to nutrient availability, the intestine must have the ability to couple nutrient sensing with hormone secretion into circulation. To investigate whether EEC production or release of AstC is regulated by nutrients, we performed immunostaining of midguts from animals following feeding or nutrient deprivation. After 6 h of starvation, AstC immunoreactivity was strongly reduced in EECs of adult female middle midguts (Fig. 1d, e). To exclude the possibility that this observation was due to reduced AstC production by the EECs, we performed gene expression analysis. *AstC* transcript levels were unchanged in adult midguts following 6 h of starvation (Supplementary Fig. 1d), suggesting that the depletion of AstC levels in the EECs in response to starvation can be attributed to increased AstC release and not to decreased production. We also confirmed that *AstC* expression and AstC immunoreactivity in the central nervous system (CNS; brain and ventral nerve cord, VNC) were unaffected by this stress (Supplementary Fig. 1d, e). Together, this indicates that AstC is released by EECs in response to nutrient restriction.

Activation of EECs results in elevation of their intracellular calcium levels[24]. We asked whether AstC-positive EECs of the adult female midgut are activated by nutrient restriction. First, we examined the expression pattern of *AstC-GAL4* (*AstC>*), a CRISPR knock-in of T2A::GAL4 at the C terminus of *AstC*[25], in the female midgut by driving *GFP* expression. *AstC>GFP* reporter activity was prominent in a large number of EECs of the middle (R3) and posterior (R4) regions and in a smaller number in the anterior R2 region (Fig. 1f). Previous work has shown that AstC is also produced in a subset of neurons in the adult brain[26]. We, therefore, combined the *AstC>* line with pan-neuronal *R57C10-GAL80* (*R57C10-G80*),

which uses an 872-base-pair enhancer fragment from *nSyb* to drive codon-optimized *GAL80* with an expression-boosting 3′ UTR in all mature, functional neurons[27–32], to suppress neuronal GAL4 and thus to restrict activity of the *AstC>* driver to the gut. Gut-specific knockdown of *AstC* using this *R57C10-G80, AstC>* driver (*R57C10-G80, AstC>AstC-RNAi*) led to highly increased starvation resistance (Fig. 1g and Supplementary Fig. 1f), confirming the effects obtained from EEC knockdown of *AstC* with the *voilà>* driver and together suggesting that gut-derived AstC is responsible for the observed phenotype. To measure the activity of AstC-positive EECs, we used the CaLexA reporter system to record intracellular calcium changes[33]. Starvation treatment for 6 h increased calcium signaling in AstC-positive EECs as indicated by a larger number of GFP-positive AstC-expressing cells as well as increased GFP intensity in these cells (Fig. 1h, i, and Supplementary Fig. 1g). This suggests that at least a subset of AstC-expressing EECs are activated when food is absent, consistent with the peptide's being released during these conditions. We next sought to determine whether EECs sense and respond directly to levels of nutrients and relay availability systemically through release of AstC. We thus acutely inhibited in the EECs, the activity of the Target of Rapamycin (TOR) pathway, a primary mediator of cell-autonomous nutrient sensing[8], by overexpressing the TOR-inhibitory proteins Tuberous Sclerosis Complex 1 and 2 (TSC1/2)[34]. This manipulation inhibits the TOR pathway in the EECs, resulting in cellular responses characteristic of nutrient deprivation[8]. After TOR inhibition was induced for 6 h using temperature-mediated GAL80 control, AstC immunoreactivity was strongly reduced in the EECs, compared to similarly treated controls, whereas intestinal *AstC* transcript levels were increased (Fig. 1j and Supplementary Fig. 1h). Next, we inhibited TOR activity specifically in AstC-producing cells of adult females using *AstC>* combined with GAL80$^{TS}$ to express *TSC1/2*. When TOR was inhibited for 6 h in the AstC-producing cells by expression of *TSC1/2*, AstC immunoreactivity was again reduced in the EECs (Fig. 1k, l), confirming the results obtained with *voilà>*-driven *TSC1/2* (Fig. 1j), without influencing CNS AstC levels (Supplementary Fig. 1i, j). These findings suggest that inhibition of TOR in the AstC-producing EECs is involved in the production and release of AstC into circulation by the intestine. Taken together our results suggest that intestinal TOR-mediated nutrient sensing in the EECs is involved in linking low nutrient availability to the release of AstC.

**AstC signaling from the gut regulates metabolic adaptations to nutritional challenges.** When animals, including *Drosophila*, encounter nutritional deprivation they must adapt by mobilizing stored energy for physiological and behavioral responses[1]. Their capacity to survive periods of nutrient scarcity is therefore directly correlated with their ability to make available stored body fat and carbohydrate resources. Whereas carbohydrates are stored mainly as glycogen and provide a fast-responding energy buffer, lipids are stored in the form of triglycerides (TAGs) in fat tissue and make up the main energy reserve. Thus, animals with increased levels of TAGs, or a reduced (but sufficient for survival) rate of their mobilization, are resistant to starvation. Given the effects of AstC on starvation survival, we investigated the physiological impact of *AstC* knockdown on flies' ability to access their energy resources. Consistent with their increased resistance to starvation (Fig. 1b, c, g), we found that females lacking *AstC* in the EECs (*voilà>AstC-RNAi*) deplete both their stored glycogen and lipids more slowly than controls (*voilà>*). Whole-body glycogen levels were higher in females with *AstC* knockdown in the EECs after 15 h starvation (Fig. 2a). Furthermore, females with EEC-specific *AstC* knockdown have higher whole-body TAG levels after 15 and 30 h starvation compared to controls (Fig. 2b, c) or the RNAi line alone (Supplementary Fig. 1k), indicating that

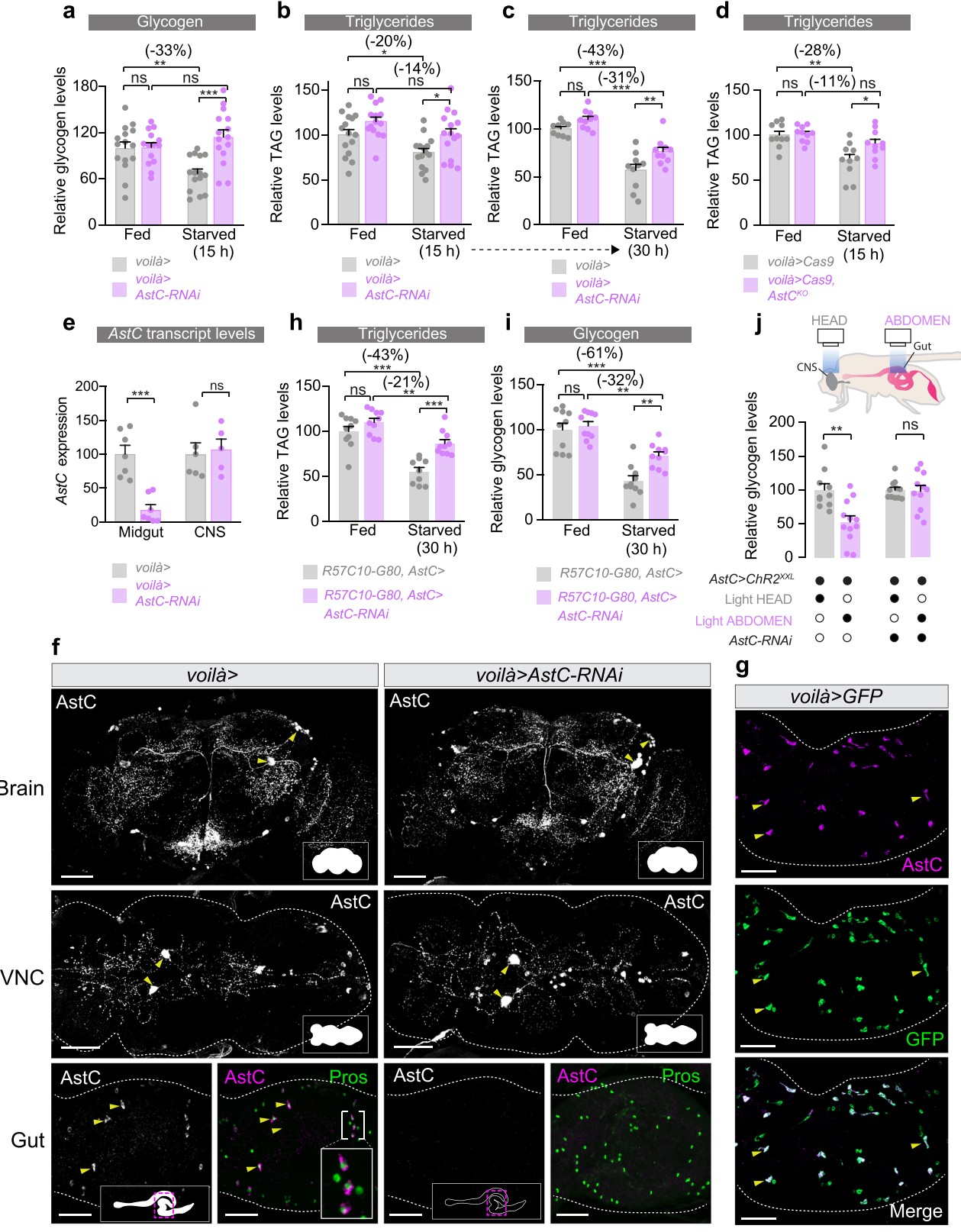

they have reduced rate of TAG depletion. When *AstC* was deleted in the adult EECs using CRISPR-mediated disruption of the gene (*voilà>Cas9, AstC^{KO}*), females also had higher TAG levels after starvation, indicating that their TAG stores are depleted more slowly than controls (*voilà>Cas9*) upon starvation (Fig. 2d), further supporting the finding that lack of *AstC* in the EECs causes a deficiency in mobilizing stored energy. These effects of

AstC on energy stores were not observed in males (Supplementary Fig. 1l, m), consistent with female-specific effects on starvation, suggesting that gut-derived AstC plays a sexually dimorphic role in metabolic regulation.

To attribute the observed phenotypes of *AstC* knockdown unambiguously to EEC depletion of AstC, we analyzed *AstC* transcript levels in dissected adult midguts and CNS. We found a

**Fig. 2 Gut-derived AstC is required for energy mobilization in response to nutritional challenges. a** Adult-female-EEC-restricted *AstC* RNAi prevents mobilization of stored glycogen during fifteen hours' starvation (column 1 vs. 3, $p = 0.0025$; column 3 vs. 4, $p < 0.0001$; two-way ANOVA genotype/diet interaction, $p < 0.01$). **b** This manipulation also reduces TAG mobilization during fifteen-hour starvation (column 1 vs. 3, $p = 0.029$; column 3 vs. 4, $p = 0.028$). (**a**, **b**, $n = 15$ except $n = 14$ for fed *voilà>AstC-RNAi*) **c** This phenotype is also displayed with longer (30-h) starvation (column 1 vs. 3, $p < 0.0001$; column 2 vs. 4, $p < 0.0001$; column 3 vs. 4, $p = 0.0056$; $n = 10$ samples except $n = 9$ for fed *voilà>*). **d** Adult-restricted CRISPR knockout of *AstC* phenocopies *AstC* knockdown (column 1 vs. 3, $p = 0.0010$; column 3 vs. 4, $p = 0.045$; $n = 10$). **e** *voilà>AstC-RNAi* depletes midgut *AstC* transcripts ($p = 0.0001$) without affecting neuronal *AstC* expression. CNS: central nervous system, consisting of brain and ventral nerve cord (VNC). $n = 7$ *voilà>* and *voilà>AstC-RNAi* midgut samples, $n = 6$ *voilà>* CNS samples, $n = 5$ *voilà>AstC-RNAi* CNS samples. **f** Brain and VNC AstC immunostaining is not obviously reduced by *voilà>AstC-RNAi*, whereas AstC staining in R3 EECs (marked with anti-Prospero) is strongly reduced. CNS staining is quantified in Supplementary Fig. 1o. Images representative of five CNSes and six guts. Scale bars 50 μm. **g** *voilà>GFP* expression in female R3 shows overlap with AstC-positive EECs. Images are representative of five guts. Scale bars 50 μm. **h**, **i** RNAi against *AstC* specifically in the AstC-producing EECs (without brain expression) reduces energy mobilization in adult females under starvation (two-way ANOVA genotype/diet interaction, $p < 0.05$; **h**: column 1 vs. 3, $p < 0.0001$; column 2 vs. 4, $p = 0.0060$; column 3 vs. 4, $p = 0.0003$; all $n = 10$ samples except $n = 9$ for starved *R57C10-G80, AstC*; **i**: column 1 vs. 3, $p < 0.0001$; column 2 vs. 4, $p = 0.0010$; column 3 vs. 4, $p = 0.0059$; all $n = 10$). **j** Blue (475 nm) illumination of abdomens, but not heads, leads to reduced glycogen levels in *AstC>ChR2^{XXL}* animals, an effect abrogated by loss of *AstC* (*AstC>ChR2^{XXL}, AstC-RNAi*): ChR2^{XXL}-induced AstC release from the gut promotes glycogen mobilization. ($p = 0.0026$; $n = 10$ head-lit *AstC>ChR2^{XXL}*, $n = 12$ abdomen-lit *AstC>ChR2^{XXL}*, $n = 11$ for both regions in *AstC>ChR2^{XXL}, AstC-RNAi*). Error bars indicate SEM. ns, non-significant; *$p < 0.05$; **$p < 0.01$; and ***$p < 0.001$. Statistical significance was determined using two-way ANOVA with Bonferroni's post hoc test (**a–d**, **h**, **i**) or two-tailed Student's *t*-test or two-tailed Mann–Whitney *U* test (**e**, **j**). Source data are provided as a Source data file.

---

strong reduction of *AstC* transcripts in midguts from *voilà>AstC-RNAi* animals, while *AstC* expression was unaltered in the CNS of these animals (Fig. 2e). We also confirmed the midgut-specific disruption of *AstC* mediated by CRISPR knockout using the *voilà>* driver (Supplementary Fig. 1n). We next performed anti-AstC immunostaining of adult CNS and midguts and found normal AstC signal levels in the brains and VNCs of *voilà>AstC-RNAi* animals, whereas AstC was undetectable in the EECs of these animals (Fig. 2f and Supplementary Fig. 1o). To further demonstrate the specificity of *AstC* knockdown to the gut, we expressed *GFP* under the control of *voilà>* to visualize expression in the gut and CNS. We found that *voilà>GFP* overlaps with AstC immunostaining in the EECs of the R3 midgut region (Fig. 2g), but not in AstC-immunoreactive neurons in the brain (Supplementary Fig. 2) or VNC (Supplementary Fig. 3). Thus, these results confirm that even though the *voilà>* driver exhibits some CNS expression, it nevertheless permits AstC-specific manipulations of the AstC-producing EECs without affecting AstC in the nervous system.

To further demonstrate that gut-derived AstC is responsible for the observed phenotypes and to exclude effects from the nervous system, we used pan-neuronal *R57C10-GAL4* to drive *AstC* knockdown in the nervous system (*R57C10>AstC-RNAi*). This led to strong reduction in *AstC* expression in the CNS, without altering midgut *AstC* transcript levels (Supplementary Fig. 4a), suggesting that the effect of *R57C10>AstC-RNAi* is specific to the nervous system. We further ruled out EEC expression of the *R57C10>* element by using it to drive *GFP* and analyzing the immunolocalization of AstC and GFP in the adult female midgut. Consistent with the nervous system-specific knockdown of *AstC*, *R57C10*-driven GFP expression was apparent in neuronal processes innervating the posterior midgut but not in AstC-positive EECs or any other cells in the midgut (Supplementary Fig. 4b). Together this shows *R57C10>AstC-RNAi* specifically targets AstC in the nervous system and not the EECs. We then examined whether neuronal knockdown of *AstC* might contribute to the phenotypes produced by *voilà>*-driven knockdown. While EEC knockdown of AstC strongly enhanced starvation resistance (Fig. 1b, c, g), neuronal knockdown did not increase starvation resistance (Supplementary Fig. 4c). Animals with neuronal loss of *AstC* also stored similar levels of TAG and glycogen and did not retain increased levels of these energy sources after starvation,

indicating that they mobilize stored energy at rates comparable to controls (Supplementary Fig. 4d, e).

AstC is expressed in both neurons and EECs, and consistently we found that *AstC>* drives efficient knockdown of *AstC* in both the CNS and the midgut (Supplementary Fig. 5a), suggesting that this GAL4 driver broadly targets the AstC-positive cell population. We suppressed the neuronal components of this driver's expression pattern by combining it with *R57C10-G80*, and we used this system to specifically reduce expression of *AstC* in the AstC-expressing EECs. *AstC* loss in the adult female AstC-producing EECs (*R57C10-G80, AstC>AstC-RNAi*) led to increased levels of TAG and glycogen after starvation, compared to controls (*R57C10-G80, AstC>*) or the RNAi line alone (Fig. 2h, i and Supplementary Fig. 5b, c). Recently, AstC-producing neurons in the brain were found to influence circadian oogenesis in adult females[35], and we, therefore, sought to determine whether gut-derived AstC has any impact upon female reproduction. Knockdown of *AstC* specifically in the EECs did not influence female egg laying (Supplementary Fig. 5d), indicating that gut-derived AstC is not important for fecundity. Together our results show that knockdown of *AstC* in the EECs using two different drivers (*voilà>* and *R57C10-G80, AstC>*) that specifically reduce expression of *AstC* in the adult female midgut —not the CNS— increases starvation resistance and reduces mobilization of stored energy during nutritional stress. On the other hand, neuronal loss of *AstC* had no effect on these phenotypes, suggesting that the phenotypes are mediated specifically by gut-derived AstC. This demonstrates the tissue specificity of our *AstC* knockdown and knockout treatments and thus identifies the adult female midgut EECs as the source of metabolism-governing AstC.

To further demonstrate that the gut, not the brain, is the source of energy-mobilizing AstC, we employed optogenetics to permit AstC-cell activation in specific tissues. We used *AstC>* to express the light-activated ion channel *Channelrhodopsin-2-XXL* (*ChR2^{XXL}*)[36] in the AstC-expressing cells of the CNS and the gut. Illumination of specific body parts—the head (brain) or abdomen (gut)—should induce the activation of the AstC-producing cells within these parts alone and thus promote AstC release. We induced activation by shining blue light (4 s per animal, every 40 s for 4 min) specifically on either heads or abdomens (harboring the AstC-producing cells in the R3 middle midgut) of immobilized adult females every 30 min for three

hours using an automated microscope stage programmed with a time series to deliver precise light pulses. We measured the influence of brain or gut AstC-cell activation on stored glycogen, which is a fast-responding source of energy, and found that $AstC>ChR2^{XXL}$ animals illuminated across their abdomens retained markedly lower glycogen compared to head-illuminated animals (Fig. 2j). Simultaneous knockdown of AstC abrogated this response, showing that this effect is mediated by light-induced release of AstC, not of any other peptides that may be co-expressed in these cells. Our findings suggest that optogenetic activation of AstC release from the gut, not the brain, causes mobilization of stored energy. Taken together, these results identify a nutrient-responsive role of AstC signaling—specifically, peptide released from the midgut EECs, without any neuronally derived peptide—that is necessary for metabolic homeostasis and adaptation to nutritional deprivation.

**AstC/AstC-R2 signaling regulates metabolic homeostasis through AKH.** In *Drosophila*, nutrient homeostasis and energy mobilization are regulated by the antagonistic actions of two key metabolic hormones, the glucagon-like factor AKH and the insulin-like DILPs[1,8]. These signaling molecules regulate metabolic pathways that govern the storage and usage of energy, which is stored mainly as glycogen and TAGs in both flies and mammals. To assess the possible involvement of insulin and AKH signaling in mediating the effects of AstC on metabolic adaptation to nutritional challenges, we examined the expression of the two AstC receptors, AstC-R1 and AstC-R2, in the IPCs and APCs to gain insight into whether AstC might act directly on these two hormonal systems. To visualize receptor localization, we used transgenic lines in which CRISPR-mediated insertion of T2A::GAL4 into the native AstC-R1 and AstC-R2 loci allows UAS-GFP reporter-based analysis of AstC-R1 and AstC-R2 expression patterns[37]. We found that AstC-R2, but not -R1, is expressed in the APCs, as indicated by co-localization with anti-AKH immunofluorescence (Fig. 3a and Supplementary Fig. 6a). Similarly, AstC-R2, but not -R1, is expressed in the adult IPCs, as indicated by overlap with anti-DILP2 signal (Supplementary Fig. 6b).

To assess the potential contribution of AKH and DILP signaling to the metabolic phenotypes associated with AstC loss in the EECs, we used RNAi to knock down AstC-R2 in the APCs or IPCs. Since loss of AKH signaling, like loss of gut-derived AstC, leads to increased resistance to starvation, we hypothesized that AKH might mediate the metabolic effects of AstC. To test this, we knocked AstC-R2 down in the adult APCs and measured organismal resistance to nutritional deprivation. APC-specific knockdown of AstC-R2 using AKH-GAL4 (AKH>AstC-R2-RNAi), restricted to the adult stage with GAL80TS (together referred to as AKH>), led to a marked increase in resistance to starvation in females with one UAS-AstC-R2-RNAi line (referred to as AstC-R2-RNAi#1), but not males, compared to controls (AKH>) phenocopying the effects of AstC knockdown in the EECs (Fig. 3b and Supplementary Fig. 6c). We also excluded possible contributory effects of the AstC-R2-RNAi transgene alone (Supplementary Fig. 6d). In contrast, adult-specific knockdown of AstC-R1 in the APCs did not prolong survival under starvation stress (Supplementary Fig. 6e), ruling out a large contribution of this receptor to AstC-mediated effects on the APCs. Furthermore, neuronal knockdown of AstC-R2 (R57C10>AstC-R2-RNAi) did not increase starvation resistance (Supplementary Fig. 4c), suggesting that AstC-R2-mediated AstC signaling in the APCs, and not the nervous system, is responsible for this phenotype. We also knocked AstC-R1 and AstC-R2 down in the corpora allata (CA), the source of juvenile hormone (JH), since AstC has been

shown to affect production of juvenile hormone in some insects[38]. Adult-restricted knockdown of AstC-R1 or AstC-R2 using the CA-specific Aug21-GAL4 driver[39] with GAL80TS (together here referred to as Aug21>) did not affect starvation resistance (Supplementary Fig. 6f), indicating that AstC regulates metabolic responses to nutrient deprivation through JH-independent means. Together this suggests that AstC signaling from midgut EECs through AstC-R2 in the APCs is a main pathway by which AstC regulates resistance to nutritional stress. We confirmed the starvation-resistance phenotype of AstC-R2 knockdown in the APCs using another independent RNAi line, whose effectiveness was increased through the co-expression of the RNA-processing enzyme Dicer-2 (Dcr-2) (Fig. 3b), as well as adult-APC-specific CRISPR/Cas9-mediated knockout of AstC-R2 (Supplementary Fig. 6g) driven by GAL80TS, AKH>Cas9. We next measured AstC-R2 transcript levels in dissected APCs from adult females and confirmed that both RNAi-mediated knockdown and CRISPR-mediated knockout efficiently reduced expression of AstC-R2 in the APCs (Supplementary Fig. 6h). This indicates that the phenotype is caused specifically by loss of AstC-R2 in the APCs and suggests that loss of AstC signaling in the APCs contributes to the increased survival of starved animals with EEC-specific AstC depletion.

The metabolic stress experienced during nutritional deprivation elicits hormonal responses that mobilize stored resources, redistribute energy towards essential physiological functions, and initiate food-seeking behaviors[1]. AKH secretion induced by hypoglycemia promotes mobilization of TAG from the fat body, and AKH-deficient animals are therefore more resistant to starvation due to their slower mobilization of TAG[40]. We also assessed resistance to starvation in animals with APC-specific RNAi against AKH (Fig. 3b and Supplementary Fig. 6d), confirming previous work showing that loss of AKH signaling leads to increased starvation resistance[40], thereby mimicking disruption of AstC signaling. Taken together, our results show that loss of AstC in the EECs or of AstC-R2 in the APCs leads to prolonged survival under starvation, implying that AstC signals to the APCs during nutritional restriction to promote systemic energy mobilization in adult females. To determine whether AstC signaling acts on the APCs to promote energy mobilization by peripheral tissues, we analyzed whole-body TAG and glycogen levels of animals with AstC-R2 depletion in the APCs to assess their energy stores. Adult-restricted AstC-R2 knockdown in the APCs led to higher whole-body TAG levels after starvation in females compared to controls (Supplementary Fig. 6i), similar to the effects observed with knockdown of AstC in the EECs, indicating a post-developmental role of AstC-R2 in the regulation of APC activity. We excluded the possibility that the RNAi line was responsible for these effects (Supplementary Fig. 6j) and obtained independent validation of these results with the second AstC-R2-RNAi line (AstC-R2-RNAi#2) in combination with Dcr-2 co-expression, which led to an even more severe metabolic phenotype in which animals depleted their stored TAGs and glycogen significantly more slowly in response to starvation (Fig. 3c, d). The effect of AstC-R2 RNAi was similar to the phenotype observed in animals with loss of AKH signaling by APC-specific knockdown of AKH, which led to a reduced depletion rate of TAGs and glycogen during nutritional deprivation (Fig. 3e, f, and Supplementary Fig. 6k). Together these findings suggest that AstC promotes lipid and carbohydrate mobilization by direct action on the APCs to promote the release of AKH during nutritional restriction.

To determine whether AKH signaling mediates the effects of AstC on resistance to starvation and metabolic stress, we assessed whether simultaneous AKH overexpression might rescue these phenotypes. Co-expression of AKH along with AstC-R2-RNAi

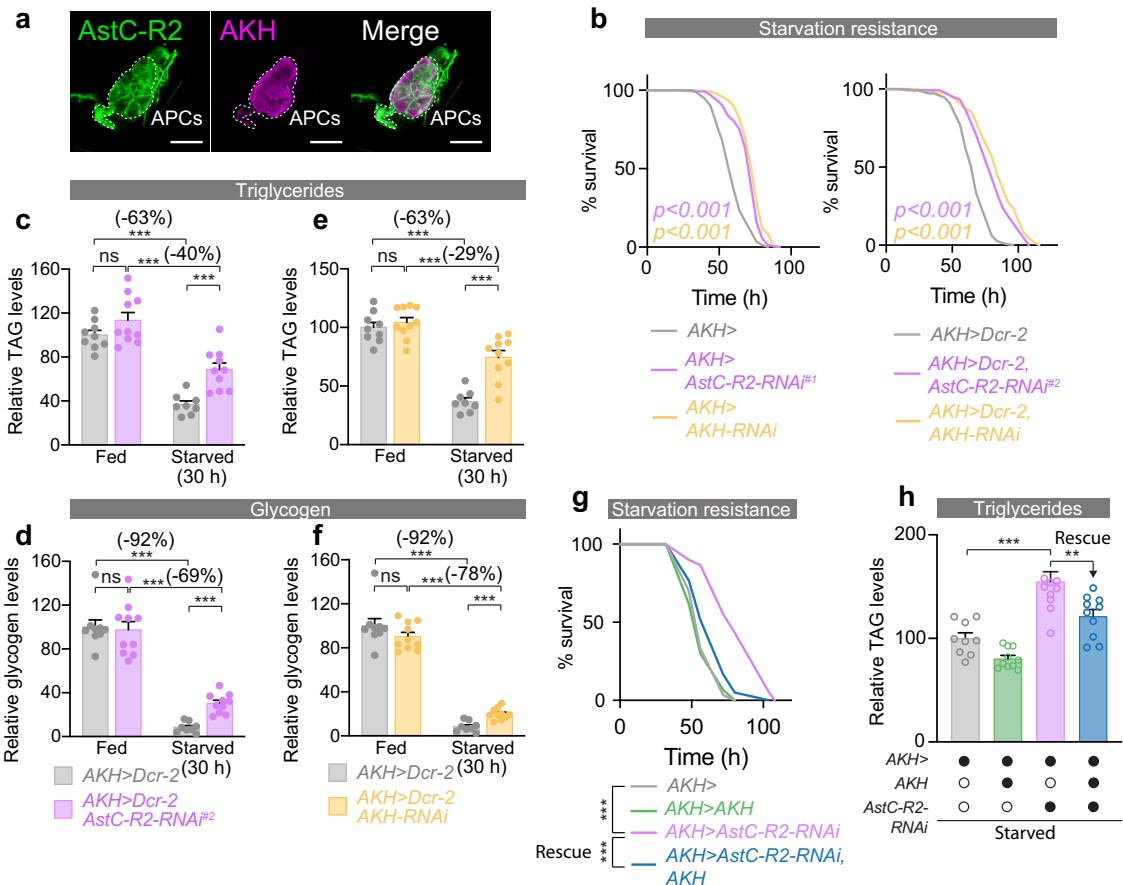

**Fig. 3 AstC acts via AstC-R2 in the APCs to regulate AKH signaling, which mediates the metabolic effects of AstC signaling. a** *AstC-R2::2A::GAL4*-driven *UAS-GFP* expression (left, green) is apparent in the AKH-producing cells (APCs) of the CC, marked with anti-AKH (magenta). Images are representative of 8 tissues. Scale bars 20 μm. **b** Knockdown of *AstC-R2* in the APCs phenocopies the starvation-survival effect of *AstC* knockdown in the gut; knockdown of *AKH* in these cells also reproduces this phenotype (*p* < 0.0001 by Kaplan–Meier log-rank test). **c, d** APC-specific knockdown of *AstC-R2* reduces the consumption of TAGs (**c**) and glycogen (**d**) during starvation (two-way ANOVA genotype/diet interaction, *p* < 0.05; **c**: column 1 vs. 3, *p* < 0.0001; column 2 vs. 4, *p* < 0.0001; column 3 vs. 4, *p* = 0.0009; **d**: column 1 vs. 3, *p* < 0.0001; column 2 vs. 4, *p* < 0.0001; column 3 vs. 4, *p* < 0.0001). **e, f** Likewise, APC knockdown of *AKH* itself induces similar reductions in starvation-induced mobilization of TAGs (**e**) and glycogen (**f**). (**e**: column 1 vs. 3, *p* < 0.0001; column 2 vs. 4, *p* = 0.0002; column 3 vs. 4, *p* < 0.0001; **f**: column 1 vs. 3, *p* < 0.0001; column 2 vs. 4, *p* = 0.0002; column 3 vs. 4, *p* < 0.0001; all *n* = 10 except *n* = 9 for fed *AKH>Dcr-2*, *n* = 8 for starved *AKH>Dcr-2*. **g** The effects of AstC signaling are mediated by AKH: overexpressing *AKH* restores normal starvation sensitivity in animals with APC-specific knockdown of *AstC-R2*. **h** Likewise, overexpression of *AKH* in animals with APC-specific knockdown of *AstC-R2* rescues their deficiency in lipid mobilization during starvation (column 1 vs. 3, *p* < 0.0001; column 3 vs. 4, *p* = 0.0055; all *n* = 10 except *n* = 9 for *AKH* > ). Error bars indicate standard error of the mean (SEM). ns, non-significant; and ***p < 0.001. Statistical significance was determined using Kaplan–Meier log-rank tests (**b**, **g**) or two-way ANOVA with Bonferroni's post hoc test (**c–f**) or one-way ANOVA with Kruskal–Wallis test (**h**). Source data are provided as a Source data file.

was sufficient to revert the starvation-resistance and energy-mobilization phenotypes caused by *AstC-R2* knockdown alone in the APCs (Fig. 3g, h), suggesting that impaired AKH signaling is the primary cause of the metabolic-stress phenotypes observed in *AstC*-deficient animals. Together, our findings suggest that communication between gut and APCs is mediated by AstC, which acts via AstC-R2 on the APCs to promote AKH secretion. This indicates that signaling between nutrient-responsive AstC-expressing EECs and the neuroendocrine CC is required for potentiation of AKH release during nutritional restriction.

Because AstC-R2 is also expressed in the adult female IPCs (Supplementary Fig. 6b), we also investigated the possibility that DILP signaling might mediate some of the effects of gut-derived AstC on the metabolic response to starvation. Knockdown of *AstC-R2* in the IPCs had no effect on starvation-induced depletion of TAG and glycogen in females (Supplementary Fig. 7a, b), indicating that energy mobilization promoted by EEC-derived AstC is not mediated through effects on IPC secretion of DILPs. Furthermore, we found

no difference in the levels of DILP2 or -3 peptides in the IPCs after starvation in females with *AstC* knockdown in the EECs or IPC knockdown of *AstC-R2* (Supplementary Fig. 7c, d), indicating that gut AstC does not strongly influence DILP secretion upon nutritional restriction. We also analyzed whether *AstC* loss in the EECs affects systemic insulin signaling by measuring levels of phosphorylated AKT (pAKT), a downstream component of insulin signaling[8]. Adult females with EEC-specific knockdown of *AstC* showed no change in whole-body pAKT levels compared to controls in fed conditions or after 6-h starvation (Supplementary Fig. 7e). Collectively, these data indicate that the stress resistance and metabolic phenotypes associated with EEC-specific depletion of *AstC* are mediated by AKH signaling and are not due to effects on DILP signaling.

**Gut-derived AstC promotes AKH secretion in response to nutritional challenges and prevents hypoglycemia.** AKH released in response to nutrient stress promotes the mobilization

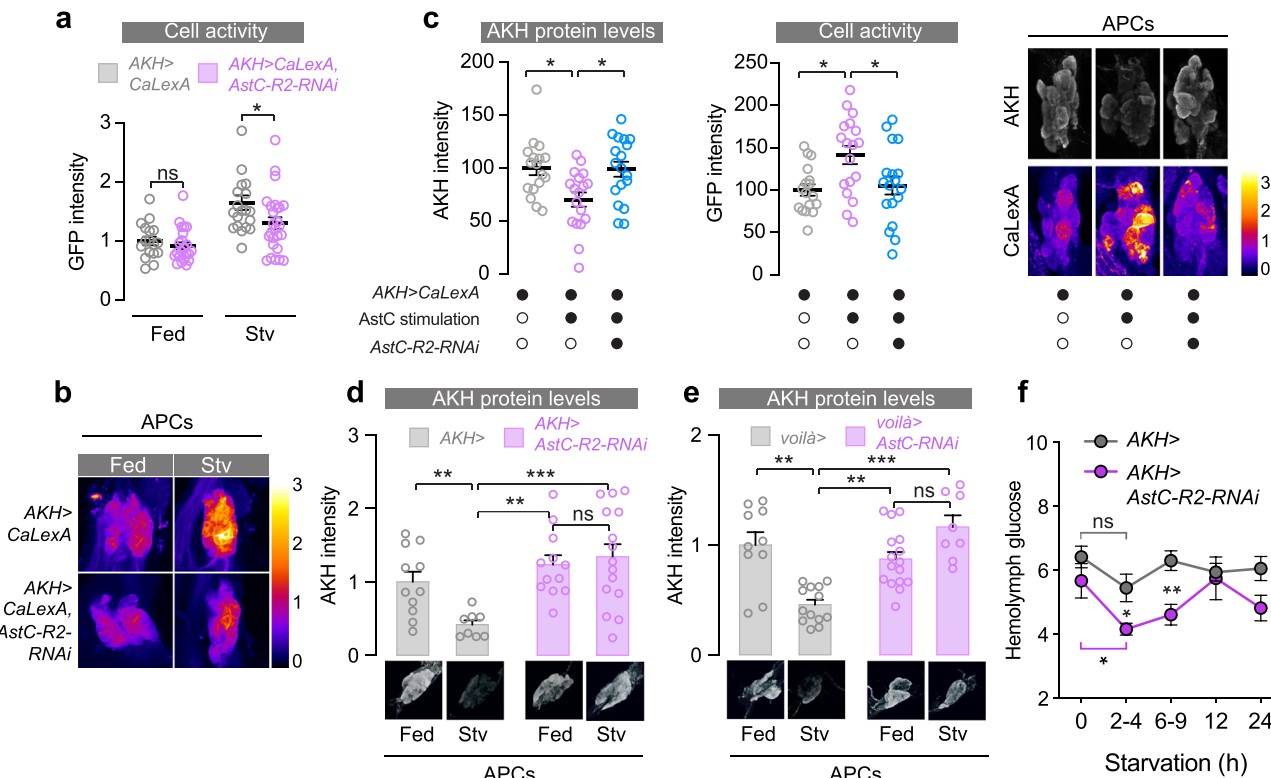

**Fig. 4 AstC acts on the APCs to promote AKH release and thereby maintain glycemic levels. a** AstC potentiates the starvation-induced activation of the APCs. Calcium-induced CaLexA-GFP in the APCs is increased by starvation in *AKH>CaLexA* control animals (gray); this increase is attenuated by *AstC-R2* knockdown ($p = 0.030$; $n = 17$, $n = 24$, $n = 20$, and $n = 26$). **b** Illustrative images of calcium-induced GFP expression in the APCs. **c** AstC application to ex vivo cultured APCs from *AKH>CaLexA* females increased CaLexA-GFP and reduced AKH staining. These effects of AstC were abolished by APC-specific knockdown of *AstC-R2* (*AKH>CaLexA, AstC-R2-RNAi*). Representative images of calcium-induced GFP expression and AKH release in the APCs are shown at right. (left panel, column 1 vs. 2, $p = 0.031$; column 2 vs. 3, $p = 0.018$; right panel, column 1 vs. 2, $p = 0.019$; column 2 vs. 3, $p = 0.042$; left to right, $n = 17$, $n = 18$, $n = 18$). **d** Immunostaining of the processed AKH peptide shows that starvation induces peptide release in *AKH>* control animals; this effect is blocked by *AstC-R2* knockdown in the APCs (column 1 vs. 2, $p = 0.0077$; column 2 vs. 3, $p = 0.0019$; column 2 vs. 4, $p = 0.0008$; left to right, $n = 11$, $n = 8$, $n = 12$, $n = 15$). See also Supplementary Fig. 8b. **e** Likewise, AstC from the gut is required for starvation-induced AKH release (column 1 vs. 2, $p = 0.0012$; column 2 vs. 3, $p = 0.0062$; column 2 vs. 4, $p < 0.0001$; left to right, $n = 10$, $n = 13$, $n = 16$, $n = 8$). Illustrative anti-AKH images of the APCs are shown below each bar. **f** AstC signaling in the APCs is required for the maintenance of circulating sugar levels in the hemolymph during starvation ($p = 0.024$; left to right, $n = 4$, $n = 8$, $n = 7$, $n = 4$, $n = 4$). Error bars indicate SEM. ns, non-significant; *$p < 0.05$; **$p < 0.01$; and ***$p < 0.001$. Statistical significance was determined using one-way ANOVA with Kruskal–Wallis test (**c**–**e**) or two-tailed Mann–Whitney *U* test (**a**, **f**). Source data are provided as a Source data file.

of resources from the fat body via its receptor AkhR to maintain energy homeostasis[10]. To further study the function of AstC in controlling AKH signaling, we examined the binding of fluorescently labeled AstC peptide to the APCs. Using this ligand-receptor binding assay[41], we found that AstC binds to the APCs in females, as indicated by co-localization of labeled AstC with *AKH>*-driven GFP fluorescence (Supplementary Fig. 8a). Furthermore, this binding of AstC was abrogated by knockdown of *AstC-R2* in the APCs, suggesting that AstC binds to AstC-R2 on the APCs.

Pancreatic α-cells respond to hypoglycemia with an increase in their intracellular calcium levels, which promotes glucagon secretion[42]. In *Drosophila*, decreased extracellular sugar concentration likewise induces increased calcium signaling in explanted APCs, suggesting that similar mechanisms regulate secretion of glucagon and AKH[43]. To test whether nutrient deprivation enhances calcium signaling in the APCs, we used the CaLexA system to report intracellular calcium concentration. Using this approach, we found that the APCs show elevated calcium-reporter activity after in vivo starvation treatment (Fig. 4a, b), indicating that APCs behave similarly to pancreatic α-cells in response to nutrient deprivation.

This starvation-induced APC calcium activity was strongly attenuated by knockdown of *AstC-R2*, demonstrating that a large part of the APC response to nutrient deprivation requires AstC signaling. To further demonstrate that AstC induces intracellular calcium activity in the APCs and thus stimulates AKH secretion in an AstC-R2 dependent manner, we stimulated ex vivo cultured APCs with AstC and measured AKH and CaLexA reporter levels. AstC stimulation of APCs decreased AKH levels, indicating increased release of AKH, which was supported by elevated calcium-reporter signal that suggested increased activity of the APCs (Fig. 4c). This response was AstC-R2 dependent, since *AstC-R2* knockdown in the APCs completely abrogated both of these effects. Taken together these findings suggest that activation of the APCs through AstC-R2 by gut-derived AstC elevates intracellular calcium levels, inducing AKH secretion and thus promoting the mobilization of stored lipids and sugars from peripheral tissues.

To assess whether AstC regulates AKH production and secretion upon nutritional challenges in vivo, we first measured *AKH* transcript levels. Consistent with starvation-induced upregulation of AKH signaling, we found that *AKH* expression was increased upon starvation (Supplementary Fig. 8b), indicat-

ing increased AKH production. This effect was abolished by knockdown of *AstC-R2* in the APCs, suggesting that AstC signaling within the APCs is required for *AKH* upregulation upon nutrient deprivation. We then asked whether AKH release during nutrient stress also requires gut AstC signaling. To test whether AstC signaling regulates the secretory activity of the APCs, we analyzed their intracellular AKH protein levels upon starvation. Starvation treatment for 6 h led to a significant drop in AKH staining levels in control females, whereas no differences in intracellular AKH levels between fed and starved conditions were observed in animals with *AstC-R2* knockdown in the APCs (Fig. 4d). This indicates that AstC-R2 is required in the APCs for starvation-induced release of AKH. We then asked whether AstC released from the gut is required for the AKH release observed upon starvation. Like removal of AstC-R2 from the APCs, EEC knockdown of *AstC* completely abolished starvation-induced AKH release, as indicated by unchanged intracellular AKH levels (Fig. 4e), suggesting that gut AstC remotely promotes AKH secretion in response to metabolic challenges. To further show that AstC is released from the gut into the hemolymph in a nutrient-dependent manner and acts systemically, we used a recently developed method[44] in which we transplanted adult female APCs into the thorax of female host animals, where they were exposed in vivo to the hemolymph of these hosts. APCs that were transplanted into pre-starved female hosts and recovered after 2 h further starvation exhibited reduced AKH staining levels compared to those transplanted into fed hosts (Supplementary Fig. 8c), indicating a humoral effect on the transplanted tissues. Next, we asked whether the hemolymph component responsible for the induction of AKH release was AstC, specifically that produced by EECs. APCs transplanted into starved female hosts carrying EEC-specific *AstC* knockout exhibited AKH levels similar to those of APCs transplanted into fed wild-type host animals. Thus, AstC is released from the EECs into the hemolymph in a nutrient-dependent manner, and this peptide is required for AKH release from the CC during nutrient deprivation.

Furthermore, receptor upregulation is a common mechanism by which cells can increase sensitivity in response to low abundance of ligand. We observed increased *AkhR* upregulation in response to starvation in females with EEC-specific loss of *AstC*, consistent with low AKH signaling (Supplementary Fig. 8d). Since nutrient deprivation increases *AKH* gene expression as described above (Supplementary Fig. 8b), these findings collectively suggest that gut-derived AstC positively modulates AKH production and release during nutrient restriction.

Glucagon acts to prevent hypoglycemia[45], and impaired AKH-induced breakdown of energy stores could result in hypoglycemia upon nutritional challenge. We, therefore, examined animals' ability to maintain normal circulating sugar levels during fasting. To assess whether AstC-mediated AKH secretion has an impact on circulating sugar concentration, we collected hemolymph before and after starvation. *AstC-R2* knockdown in the APCs led to fasting-induced hypoglycemia (Fig. 4f), consistent with an impairment of AKH-mediated energy mobilization. Control females maintained relatively constant hemolymph glycemia during 24 h of fasting, while hemolymph glucose concentration dropped significantly within a few hours of fasting when *AstC-R2* function was depleted in the APCs. Consequently, females with APC-specific *AstC-R2* knockdown exhibited reduced circulating sugar levels, indicating they were in a hypoglycemic state for several hours following starvation. These findings together indicate that AstC released by the endocrine cells of the gut in response to nutrient deprivation acts on the APCs to promote AKH release and thus to promote energy mobilization to prevent hypoglycemia.

**Gut-derived AstC promotes food intake via AKH signaling.** Homeostatic control of energy balance requires the coordination of food intake and energy expenditure[1]. Metabolic processes are therefore intimately coupled with regulation of feeding to maintain a constant internal state. To investigate whether AstC signaling coordinates metabolic and behavioral responses to nutrient deprivation, we monitored food intake in animals with adult-restricted EEC-specific *AstC* knockdown. *AstC* loss in the EECs caused a significant decrease in short-term (30-min) food intake measured by dye-consumption assay in females (Fig. 5a), an effect not caused by the RNAi line itself or observed in males (Supplementary Fig. 9a). To further understand the endocrine mechanisms by which *AstC* deficiency causes hypophagia, we tested whether AstC regulates food intake through its action upon the APCs, using adult-restricted knockdowns. Using the short-term dye-feeding assay, similar results were obtained for females when *AstC-R2* or *AKH* was knocked down in the APCs as when *AstC* was depleted in the in EECs (Fig. 5a and Supplementary Fig. 9a). We also ruled out any major contribution of neurons expressing AstC or its receptor AstC-R2 to these feeding behaviors (Supplementary Fig. 9a). Next, we quantified long-term food consumption using the capillary feeder assay (CAFÉ) assay[46] to monitor food intake over 24-h periods. Consistently, long-term food consumption was reduced in females when *AstC* was knocked down in the EECs or when *AstC-R2* or *AKH* itself was knocked down in the APCs (Fig. 5a). We then tested the impact of AstC signaling on feeding regulation using the FLIC (fly liquid-food interaction counter) system, which enables analysis of *Drosophila* feeding behaviors[47]. We found that adult-specific knockdown of *AstC* in the EECs reduced the number of tasting and feeding events while increasing the time interval between feeding events over 24 h (Fig. 5b). Thus, despite having TAG and glycogen levels similar to those of fed control animals, as shown above, fed females with reduced AstC signaling are hypophagic. This shows that the metabolic phenotype is not caused by increased food intake or impaired nutrient absorption. Given that decreasing EEC-derived AstC signaling led to a reduction in food intake, we investigated whether enhanced AstC signaling from EECs might promote feeding. This was achieved by ectopically expressing the thermosensitive *Transient receptor potential A1* (*TrpA1*) cation channel, which is activated by slightly elevated temperatures such as 29 °C[48], in the AstC-producing EECs (with R57C10-G80 to suppress neuronal GAL4 activity). Exposure to 29 °C activation conditions to induce peptide release caused a strong increase in feeding behavior compared to the 20 °C non-activation condition and *w1118* control animals, indicating an orexigenic effect of hormones released from AstC EECs (Fig. 5c). This behavioral effect was attenuated by knockout of *AstC* in the targeted cells (Fig. 5c; two-way ANOVA, $p = 0.03$), and thus by reduced TrpA1-induced peptide release, indicating that AstC release from EECs is required for feeding promotion. Together our results indicate that in response to nutrient deprivation, AstC signaling from the gut promotes food consumption through regulation of AKH, consistent with previous findings showing that AKH is an orexigenic peptide[49,50].

Organisms also respond to nutritional challenges by altering their food-seeking behavior. In *Drosophila*, starvation induces hyperactivity and suppresses sleep, presumably to promote food-seeking activities. Since AKH is required for starvation-induced hyperactivity and sleep inhibition, we investigated whether AstC is important for sleep responses to food deprivation. We monitored sleep under normal feeding and for 24 h following nutrient deprivation in animals with *AstC* knockout in the EECs. Under fed conditions, control and knockout animals displayed similar sleep patterns during the night (Fig. 5d, e, and Supplementary Fig. 9b). Under starvation, however, while

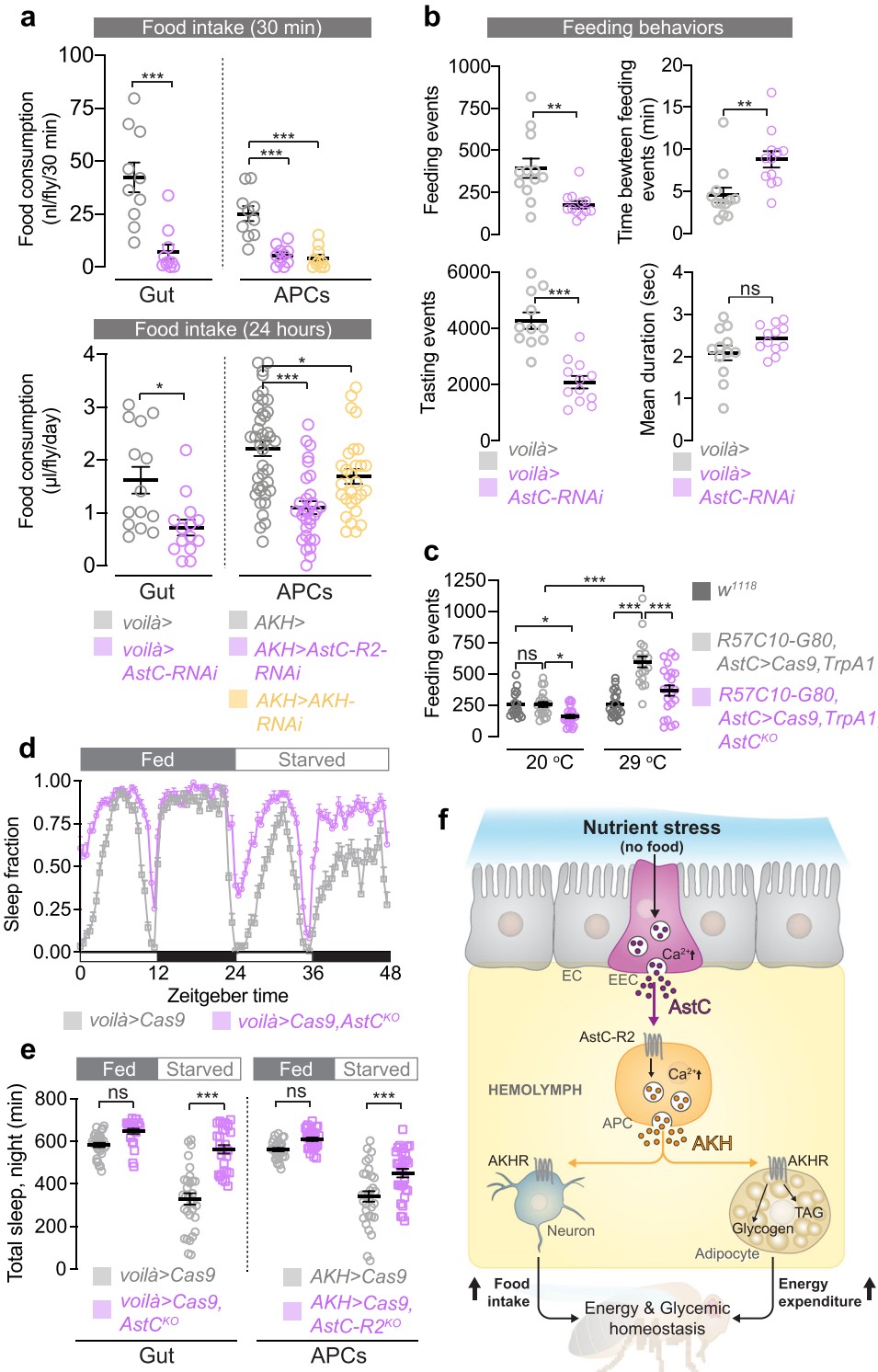

controls exhibited a strong reduction in night sleep, females with EEC-specific *AstC* deletion showed almost no sleep loss (Fig. 5d, e). Sleep and feeding are functionally interconnected behaviors that are mutually exclusive[51]. Our results indicate that animals with *AstC* knockdown in the gut, even in fed conditions, sleep more during the day compared to controls (Supplementary Fig. 9b). To gain insight into whether the reduced food intake in animals lacking gut *AstC* is a consequence of increased sleep, we binned daytime sleep and feeding episodes into three four-hour intervals. During the middle of the day (ZT 4–8), when female

controls and females with EEC-specific *AstC* knockdown exhibit similar amounts of sleep (Supplementary Fig. 9c, middle pair), knockdown animals still exhibit reduced feeding behaviors (Supplementary Fig. 9d), indicating that AstC affects food intake, at least in part, through a mechanism that is independent of sleep regulation. Next, we tested whether AstC regulates sleep loss during starvation via AstC-R2 in the APCs. As seen with knockout of *AstC* in the EECs, females with *AstC-R2* knockout in the APCs resisted, at least partially, the starvation-induced sleep loss (Fig. 5e and Supplementary 9e) displayed by controls. These

**Fig. 5 AstC promotes food-seeking behaviors. a** *AstC* knockdown in EECs or *AstC-R2* or *AKH* knockdown in APCs reduces feeding in dye assays (top: gut $p = 0.0002$; APCs column 1 vs. 2, $p < 0.0001$; column 1 vs. 3, $p < 0.0001$; all $n = 10$) and in longer-term CAFE assays (bottom: gut $p = 0.029$; APCs column 1 vs. 2, $p < 0.0001$; column 1 vs. 3, $p = 0.016$; all $n = 10$). **b** EEC *AstC* knockdown reduces feeding and tasting events recorded by the FLIC system and increases the interval between feeding events; the duration of individual events is unaltered (feeding events, $p = 0.0022$; time between events, $p = 0.0023$; tasting events, $p < 0.0001$; all $n = 12$ except $n = 11$ for *voilà>* tasting events). **c** TrpA1-mediated activation of AstC-producing EECs leads to increased feeding behaviors, an effect diminished by *AstC* knockout (column 1 vs. 3, $p = 0.042$; column 2 vs. 3, $p = 0.022$; column 2 vs. 5, $p < 0.0001$; column 4 vs. 5, $p < 0.0001$, column 5 vs. 6, $p < 0.0001$; two-way ANOVA genotype/temperature interaction, $p < 0.05$. Left to right, $n = 17$, $n = 21$, $n = 23$; $n = 20$, $n = 19$, $n = 22$.). Full genotype: *R57C10-GAL80; AstC::T2A::GAL4/UAS-TrpA1, UAS-GAL4; tub-GAL80^{ts}, UAS-Cas9*, with or without *UAS-AstC^{KO}* on the third chromosome. Because KO disrupts the *AstC::T2A::GAL4* locus, *UAS-GAL4* continues GAL4 expression after this event. **d** EEC knockout of *AstC* does not strongly affect sleep architecture in fed females; starvation suppresses sleep, an effect attenuated by *AstC* knockout. **e** Starvation suppresses nighttime sleep in *voilà>Cas9* and *AKH>Cas9* animals, an effect attenuated by *AstC* KO in the EECs or *AstC-R2* KO in the APCs. (gut, $p < 0.0001$; APCs, $p < 0.0001$; left to right in **e**: $n = 31$ animals, $n = 24$, $n = 31$, $n = 28$; $n = 31$, $n = 31$, $n = 31$, $n = 31$. N values apply to **d** also). **f** A model of AstC-mediated nutritional signaling from the gut. Reduced nutritional levels induce the EECs to release AstC into the hemolymph. Although the APCs can autonomously sense circulating sugar levels, they also require AstC input via AstC-R2 to potentiate their activity during starvation, promoting AKH expression and release into the hemolymph. AKH acts via AkhR (1) on the fat body to induce the mobilization of stored energy and (2) on neurons modulating feeding and arousal to suppress sleep, upregulate locomotion, and promote feeding behaviors. Through these effects, gut-derived AstC promotes energetic and nutritional homeostasis. Error bars indicate standard error of the mean (SEM). ns, non-significant; *$p < 0.05$; **$p < 0.01$; and ***$p < 0.001$. Statistical significance was determined using two-way ANOVA with Bonferroni's post hoc test (**c**), one-way ANOVA with Dunnett's test or with Kruskal–Wallis test (**a**), or two-tailed Mann–Whitney *U* test (**a**, **b**, **e**). Source data are provided as a Source data file.

results indicate that AstC, via the APCs, regulates behavioral responses to starvation in order to initiate food-seeking. Combined, our findings show that gut-derived AstC controls feeding by AstC-R2-mediated effects in the APCs that potentiate hypoglycemia-induced AKH secretion. This indicates that under conditions of nutrient restriction, AstC signaling promotes food-seeking behaviors and therefore enhances food consumption to coordinate the organism's homeostatic regulation of food intake and energy usage.

## Discussion

The gut has emerged as a central signal-integrating regulator of whole-body energy homeostasis. Despite the evolutionary divergence of flies and humans, the guts of these animals exhibit physiological and structural similarity, with the *Drosophila* midgut functioning analogously to the human small intestine[6,7]. Work in mammals and *Drosophila* has shown that the gut integrates nutritional and microbiotic cues and converts these into hormonal signals that relay nutritional information systemically to other tissues. However, our understanding of the cellular and molecular bases of nutrient sensing in the EECs, as well as the hormones these cells release, is limited. We conducted a screen of EEC-expressed peptide hormones for effects on starvation survival. Among our strongest survival-prolonging hits was AstC, one of the most highly and broadly expressed peptides in the *Drosophila* midgut[52,53]. Despite its abundance, the function of gut-derived AstC has remained unknown. Here, we show that certain midgut EECs respond to nutrient deprivation by secreting AstC into the hemolymph, which induces energy mobilization and promote food-seeking behaviors (Fig. 5f). We show that although AstC is present within the nervous system, this hormonal pool does not contribute to the observed nutritional-homeostatic effects of AstC signaling. EEC retention of AstC under fed conditions requires the activity of the intracellular nutrient sensor TOR, which is promoted by nutrients, particularly amino acids[8]. EEC-derived AstC acts directly upon the APCs, which express the receptor AstC-R2, to promote AKH release. Playing a role analogous to that of mammalian glucagon, AKH acts on its receptor AkhR in adipose tissue to induce breakdown of energy stores, thereby protecting against deleterious drops in hemolymph sugar levels leading to hypoglycemia during fasting or periods of rapid energy expenditure[40].

Nutrient stress also modulates feeding decisions and behaviors to optimize survival. Nutritional deprivation increases locomotor activity and suppresses sleep in animals as an adaptive response to promote food-seeking behaviors[40,54]. AKH mediates this starvation-induced behavioral arousal response through its action on a small number of octopaminergic neurons in the brain that express AkhR[11]. Like human glucagon, AKH also has direct orexigenic effects, acting through four interoceptive AkhR-expressing neurons in the suboesophageal zone to promote consumption[50]. Our findings show that gut-derived AstC signaling is required for starvation-induced sleep suppression and feeding increase via actions that are mediated, at least in part, by AstC-R2 in the APCs. Together our results suggest that starvation induces AstC release from EECs into the hemolymph to increase appetite, arousal, and food-seeking behaviors, along with breakdown of energy stores, through regulation of AKH (Fig. 5f). Thus, AstC coordinates behavioral and metabolic responses that allow organismal adaptation to nutritionally stressful environmental conditions.

In contrast to the regulation of insulin release, which is well studied, the mechanisms governing glucagon secretion are poorly defined[55] and have recently become a focus of intensive research due to their relevance to diabetes. Likewise, in the fly, the regulation of insulin secretion has been examined intensively, whereas the mechanisms by which AKH secretion is controlled are poorly understood, although the APCs, like mammalian pancreatic α-cells, are believed to be competent to sense circulating sugar levels[43,44]. Our present findings show that animals lacking *AstC* in the EECs, or its receptor *AstC-R2* in the APCs, fail to regulate their mobilization of stored energy properly during nutrient deprivation and thus deplete their stores of TAGs and glycogen more slowly than normal; as a consequence, these animals exhibit prolonged starvation survival, at the expense of becoming hypoglycemic. Thus, AstC signaling is required for AKH secretion in response to fasting to mobilize energy and avoid hypoglycemia. Paradoxical glucagon secretion under hyperglycemia is frequently observed in diabetic patients[56]. Thus, dysregulation of signals that act similarly to AstC, promoting glucagon-like signaling, may contribute to hyperglycemia in diabetes and therefore be of therapeutic relevance.

*Drosophila* AstC-R2 is orthologous with mammalian somatostatin receptors[21], and the processed AstC peptide shares structural features with somatostatin, both being cyclized via disulfide bridges. Somatostatin (as its name suggests) is known for its regulatory effects on growth hormone, although it has recently also received attention for its role in the regulation of

glucagon secretion[57]. Two active forms of the somatostatin peptide, which elicit partially overlapping responses, are produced by different tissues. Like AstC, which is produced by a small number of neurons in the *Drosophila* brain and is also highly expressed in the fly intestine, somatostatin is produced by the hypothalamus, but the major source of circulating peptide is the gastrointestinal tract. Endocrine cells of the gut produce the long somatostatin-28 isoform[58,59], and the δ (delta) cells of the pancreas synthesize the short somatostatin-14 isoform, which has a strong paracrine intra-islet regulatory effect on glucagon secretion. However, less is understood about whether systemic somatostatin signaling from the intestine might signal to the pancreas and exert a role there in the regulation of glucagon secretion. Our results demonstrate that in *Drosophila*, the EECs release AstC systemically, allowing it to act directly upon the APCs to regulate AKH secretion, suggesting the possibility that mammalian gut-derived somatostatin may play a similar role in regulating glucagon secretion. We propose that AstC/somatostatin and AKH/glucagon define an evolutionarily conserved endocrine module regulating homeostatic control of food intake and metabolism.

In addition to processing ingested food, the intestine functions as a barrier between the external and internal environments that plays a key role in immunity, and the gut microbiome plays important roles in host physiology and health[60]. The mechanisms by which gut bacteria affect host physiology are believed to include the sensing of microbial metabolites by EECs, which respond by releasing hormones that act as cytokines[61]. Interestingly, AstC signaling mediated by AstC-R2 is also involved in *Drosophila* inflammatory and immune responses[62]. Other nutrient-sensitive gut-derived hormones appear to regulate both of these pathways as well, such as Burs signaling through its receptor Lgr2/Rickets[19,63], and the interconnected effects of gut signaling on metabolism and immunity will be an interesting area for future exploration.

Males and females differ in their metabolic physiology and susceptibility to disease[64]. These differences are likely associated with distinct male and female reproductive strategies and lifestyles. In both mammals and *Drosophila*, females store more lipids than males do[22,65]. Female flies also consume more food than males[1] and have a higher demand for dietary amino acids[66], which are required to sustain metabolically demanding oogenesis, whereas reproduction is less metabolically costly for males. Our finding of female-specific AstC function raises the possibility that this peptide may affect some of these female-specific metabolic and behavioral adaptations, contributing to greater female reproductive success. Females also must promote the developmental success of their offspring, which require dietary proteins for their development. They may therefore respond more quickly to a lack of such nutrients than males by initiating food-seeking behaviors in order to locate rich sites in which to deposit their eggs. It will be of interest for future studies to further explore the female-specific function of AstC, including whether AstC is linked to behaviors and physiology associated with female reproduction. Our observations of sex-specific differences are perfectly in line with recent work showing that sexual dimorphism in TAG metabolism in *Drosophila* differentially governs starvation resistance in males and females and is regulated by AKH[22,67]. In males, this sex-specific difference depends on the lipase Brummer that promotes lipolysis of stored TAG in response to starvation. Our results indicate a role of AstC in the regulation of sex-specific differences that underlie female-specific TAG metabolism in response to nutritional challenges, which may be important for a better understanding of sex differences in metabolism and disease therapies.

Balance between food intake and energy expenditure is necessary for the maintenance of energy homeostasis, and

imbalance in these processes can lead to obesity or wasting. Organismal energy balance therefore requires the ability to sense food cues and to regulate feeding and metabolism appropriately in response[1]. Nutrient-sensing pathways and the regulation of metabolism and feeding have become the focus of intense research, as these are the keys to controlling energy balance and are often dysregulated in human metabolic disorders[68]. Intestinal nutrient sensing has emerged as a major endocrine regulator of body-wide metabolism, food intake, and blood-glucose control, and targeting intestinal nutrient sensing and downstream gut-derived hormones holds great promise for the treatment of diabetes and obesity. The ability of the gut to sense nutrients is critical for regulating its release of hormones involved in appetite and metabolic control. Our work shows that gut AstC release is regulated by a mechanism that involves TOR-dependent nutrient sensing in the EECs, also indicating that the APCs sense nutrient status indirectly via the EECs and AstC. TOR may be acting as a permissive signal for release of AstC from EECs during nutritional stress along with other factors. Although we did not observe effects on AstC in the CNS of the nutritional stress or inhibition of TOR that affected gut AstC, it is possible that neuronal AstC is influenced by different nutritional conditions. How the brain senses nutrition and whether it involves AstC will be interesting for future studies to further examine. TOR is activated by amino acids, but glucose also promotes TOR activity by inhibiting the TOR-inhibitory energy sensor AMPK and thus disinhibiting TOR[69]. Thus, deprivation of amino acids or glucose inhibits TOR, which is a primary mechanism by which cells and organisms sense scarcity and induce adaptive growth and metabolic responses to dietary restriction. We now provide evidence that intestinal nutrient sensing controls body-wide metabolism and feeding through the release of AstC, which promotes food intake and energy expenditure.

In mammals, glucose sensing in the EECs is a main mechanism by which insulin secretion is regulated, since the combined actions of the incretin hormones GLP-1 and GIP, both released from EECs upon the ingestion of glucose, potentiate hyperglycemia-induced insulin secretion[70]. Incretin effects are impaired in diabetic patients, and clinical use of GLP-1-agonist therapies has recently become a widely used strategy for the treatment of diabetes and obesity. In these metabolic disorders, impaired intestinal nutrient sensing and incretin signaling lead to a failure to maintain glycemic homeostasis and regulate food intake[71]. We show here that AstC is a hormone secreted from intestinal EECs that induces AKH secretion, providing evidence of a gut hormone with potential incretin-like properties that acts on glucagon-like signaling systems. Taken together our findings identify an endocrine pathway that regulates homeostatic control of food intake and energy metabolism. Identification of hormonal factors such as AstC that control food intake and energy metabolism may be exploited to control appetite and body weight and help pave the way for development of new strategies for the treatment of diabetes and obesity.

## Methods

***Drosophila* stocks and husbandry.** Flies were maintained on a standard cornmeal medium (containing 82 g/L cornmeal, 60 g/L sucrose, 34 g/L yeast, 8 g/L agar, 4.8 mL/L propionic acid, and 1.6 g/L methyl-4-hydroxybenzoate) at 25 °C and 60% relative humidity under normal photoperiod (12 h light: 12 h dark). Genotypes with temperature-sensitive GAL80 (*Tub-GAL80^TS*) were raised through eclosion at 18 °C. Newly eclosed flies were kept at 18 °C for 3–4 days more prior to separation by sex (20 flies per vial) and transferred to 29 °C for 5 days to induce UAS expression before the start of experiments. Flies were flipped onto fresh medium every third day. Lines obtained from the University of Indiana Bloomington *Drosophila* Stock Center (BDSC) include the following: *Akh-GAL4* (#25684); *Ast-C::2A::GAL4* (#84595) and *AstC-R1::2A::GAL4* (#84596)[25]; *R57C10-GAL4* (#39171); CaLexA system (#66542): *LexAop-CD8::GFP::2A::CD8::GFP; UAS-Lex-A::VP16::NFAT, LexAop-CD2::GFP/TM6B, Tb*[33]; *UAS-ChR2^{XXL}* (#58374);

*Tub-GAL80^{TS}* (#7108); *UAS-Akh* (#27343); *UAS-Cas9.P2* (#58985)[72]; *UAS-TrpA1* (#26263); *UAS-Tsc1, UAS-Tsc2*[34] (#80576); and UAS-RNAi lines[73,74] targeting *AstC-R1* (#27506) and *AstC-R2* (*AstC-R2-RNAi^{#2}*, #25940). The Kyoto Stock Center provided *UAS-GAL4* (#108492). The Vienna *Drosophila* Resource Center (VDRC) provided the line *w^{1118}* (#60000, genetic background of VDRC RNAi lines), which crossed to the *GAL4* driver lines was used as a control in all experiments, as well as numerous UAS-RNAi lines[75] listed in Table S1, including lines targeting *AKH* (#105063), *AstC* (#102735), and *AstC-R2* (*AstC-R2-RNAi^{#1}*, #106146). All *GAL4* and *GAL80* lines as well as transgene combinations were standardized to carry similar genetic backgrounds using chromosomal balancer lines that carry similar genetic background to *w^{1118}*, except for the balancer chromosomes. Briefly, the balancer lines were previously crossed to our laboratory *w^{1118}* line, to introduce a genetic background population similar to the *w^{1118}*. In order to have certain small standardized genetic variations in all lines, at least 15 individuals of *w^{1118}* were used for backcrossing. Females were used to also standardize the genetic background of the X chromosome. Therefore, all transgenes combined with different transgenes were also carrying similar genetic backgrounds. Mated females were used in all experiments. *Ilp2-GAL4, UAS-GFP* (original GAL4 construct from BDSC #37516[76]) was a gift of Pierre Léopold (Institut Curie, France). *voilà-GAL4*[77] was a gift of Alessandro Scopelliti (University of Glasgow, UK). *AstC-R1::2A::GAL4*[37] and *AstC-R2::2A::GAL4*[37] were gifts of Shu Kondo (Tohoku University). *R57C10-GAL80-6 (R57C10-GAL80-WPRE[su(Hw) attP8])*[27–32] (on the X chromosome) was a gift of Ryusuke Niwa (University of Tsukuba).

**Generation of *UAS-gRNA* lines for tissue-specific CRISPR knockouts.** GAL4-inducible *UAS-2xgRNA* constructs allowing tissue-specific disruption of the *AstC* or *AstC-R2* loci were generated in the pCFD6 backbone[78], obtained from AddGene (#73915; https://addgene.org/). The target sequences (Table S2) were identified using the E-CRISP service[79] (https://e-crisp.org/E-CRISP/). The *AstC* construct should lead to deletion of most of the peptide-encoding sequence; the *AstC-R2* construct should remove a large part of the locus encoding the transmembrane domains of the receptor, even if the gene retains any function. Putative clones were sequenced, and correct constructs were integrated into the fly genome at the attP2 (third chromosome) site by BestGene, Inc. (Chino Hills, CA).

**Starvation-survival assays.** To assay survival under starvation, flies were transferred without anesthesia to vials of starvation medium (1% agar in water) and kept at 29 °C for animals with GAL80^{TS} and at 25 °C for animals without temperature control; the number of dead animals was assessed at time intervals until the completion of the assay. For each genotype, 90–150 flies were used, except for the rescue experiment with *AKH* overexpression, in which 60 flies were used. Statistical significance was computed using the Kaplan–Meier log-rank survival functions of the Prism program (GraphPad).

**Immunohistochemistry and confocal imaging.** Adult midguts and brains were dissected in cold PBS and fixed in 4% paraformaldehyde in PBS at room temperature for 45 or 60 min, respectively, with gentle agitation. For CC (APCs) preparations, the CC were slightly pulled out from the head and pre-fixed with head for 30 min to minimize variations in the starvation period between different animals and treatments. After pre-fixation, CC were finely dissected and fixed again with 4% paraformaldehyde in PBS for an additional 10 min. Tissues were rinsed with PBST (PBS + 0.1% Triton X-100, Merck #12298) three times for fifteen minutes each, blocked with 3% normal goat serum (Sigma) in PBST at room temperature for 30 minutes, and incubated overnight at 4 °C with gentle agitation in primary antibodies (below) diluted in blocking solution, except for brains, which were left for two days. Primary-antibody solution was removed, and tissues were washed three times with PBST and incubated with secondary antibodies diluted in PBST overnight at 4 °C with gentle agitation, avoiding light. Tissues were washed three times with PBST and mounted in VECTASHIELD mounting medium containing DAPI as counterstain (Vector Laboratories, #H-1200) on slides treated with poly-L-lysine (Sigma, #P8920). Tissues were imaged on a Zeiss LSM-900 confocal microscope using a 20x objective and a 1-micron Z-step in the Zen software package. Post-imaging analysis was performed using the open-source *FIJI* software package[80]. For quantification of AKH, AstC, and GFP (CaLexA reporter activity) staining intensity, stacks were Z-projected using the Sum method, and signals were measured as the Integrated Density in FIJI. For AstC and GFP (CaLexA reporter activity) in midgut, quantifications were done on the middle-midgut R3 region. For each cell or tissue the integrated intensity of an adjacent unstained area was measured, and this background was subtracted to give the net signal. Antibodies used included rabbit antibody against the processed AKH peptide[40] (kind gift of Jae Park, U. Tennessee), 1:500; mouse anti-Prospero clone MR1A[81] (University of Iowa Developmental Studies Hybridoma Bank), 1:20; rabbit anti-DILP2[82] (kind gift of Ernst Hafen, ETH Zurich), 1:1000; mouse anti-DILP3[14] (kind gift of Jan Veenstra, University of Bordeaux), 1:500; rabbit anti-AstC[14] (also kindly given by Jan Veenstra and Meet Zandawala), 1:500; mouse anti-GFP (Thermo-Fisher #A11120), 1:500; Alexa Fluor 488-conjugated goat anti-mouse (Thermo-Fisher #A32723), 1:500; Alexa Fluor 488-conjugated goat anti-rabbit (ThermoFisher #A11008), 1:500; Alexa Fluor 555-conjugated goat anti-mouse

(ThermoFisher #A32727), 1:500; and Alexa Fluor 555-conjugated goat anti-rabbit (ThermoFisher #A21428), 1:500.

**Binding assay with labeled AstC.** BODIPY-labeled AstC was synthesized by Cambridge Peptides, Ltd (UK). The endogenous AstC peptide has the sequence QVRYRQCYFNPISCF, cyclized via a disulfide bridge between its cysteines. It is not C-terminally amidated, but the N-terminal residue is processed into a pyr-oglutamate. Since the C-terminal end of the peptide is more conserved between species and thus likely most important for binding, we sought to label the N terminus. Pyroglutamate lacks a free N-terminal amino group (by which labels are linked), so it was replaced by a glycine spacer. Thus, the peptide GGVRYRQ-CYFNPISCF was synthesized and cyclized via its cysteine residues; BODIPY-TMR (BODIPY-543) dye was conjugated to the N terminus by an NHS ester linkage. Synthesized peptide was redissolved at 50 μM in 50%/50% DMSO/water. CC complexes were dissected from adult females expressing GFP in the CC (*AKH>GFP* crossed to either *w^{1118}* or *UAS-AstC-R2-RNAi #106146*) in artificial hemolymph-like solution[83] containing 60 mM sucrose (inert osmolyte) and 60 mM trehalose, which mimics a fed-like state. Tissues were mounted in artificial hemolymph on poly-lysine-coated glass-bottomed dishes (MatTek, #P35G-0-10-C) and used in a ligand-receptor binding assay as previously described[41]. In brief, labeled peptide was diluted to 1 μM in artificial hemolymph and added at a 1:10 ratio to the mounted tissues (final concentration 100 nM), and bound peptide was imaged using an LSM-900 confocal microscope (Zeiss) in AiryScan2 super-resolution mode with a 40x objective and GFP and BODIPY-543 filter settings.

**In vitro CC stimulation assay.** The APCs (CC) along with a small piece of oesophagus and proventriculus) from 5-day-old *AKH>CaLexA* or *AKH>CaLexA, AstC-R2-RNAi* adult females were dissected out in artificial hemolymph[83] containing a low, starvation-like level of trehalose (5 mM) + 115 mM inert sucrose osmolyte and pre-cultured for at least 30 min to equilibrate. Dissected CC were incubated in the same culture medium for 2 h at room temperature in the presence or absence of unlabeled synthetic AstC (final concentration, 1 μM). AstC peptide, including the N-terminal pyroglutamate (pQ) and cystine linkage (*)—pQVRYRQC*YFNPISC*F—was synthesized by ThermoFisher. Cultured CC were fixed after incubation in 4% paraformaldehyde/1xPBS solution for immunocytochemistry as described above.

**Transplantation assay.** The APCs (CC) along with a small piece of oesophagus and proventriculus from 5-day-old adult females were transplanted into 5-day old host females, either controls (*voilà>Cas9*) or gut-specific *AstC* knockouts (*voilà>Cas9, AstC^{KO}*). Four hours before transplantation, the host adults were transferred either to starvation medium or to fresh food vials, to induce (or not) starvation-promoted AstC secretion from the EECs into the host hemolymph. Host flies were anaesthetized with $CO_2$ and their wings were carefully removed. An incision was made along the border between the scutum and scutenum, and the CC-containing tissues were implanted into the anterior face of the incision (into the dorsal thorax) using fine forceps. Transplanted tissues were incubated within the host animals (continuing on the same feeding substrate as before) for two hours at 29 °C; the transplanted CC were then recovered from the hosts and fixed immediately in 4% paraformaldehyde/PBS solution for immunocytochemistry as described above.

**Transcript measurement using qPCR.** RNA isolation was performed using the RNeasy Mini Plus kit (Qiagen #74136). Tissues (five CNSs, five guts, five CC (APCs), or five whole animals per sample, times 5–8 replicate samples per condition or genotype) were homogenized in 2-mL tubes containing lysis buffer plus 1% 2-mercaptoethanol using a TissueLyser LT bead mill (Qiagen) and 5-mm stainless-steel beads (Qiagen #69989) following the manufacturer's protocol. RNA was reverse-transcribed using the High-Capacity cDNA Synthesis kit (Applied Biosystems, #4368814). QPCR was performed using RealQ Plus 2x Master Mix Green (Ampliqon, #A324402) on Stratagene Mx3005P (Agilent Technologies) and Quant Studio 5 (Applied Biosystems) machines. Results were normalized against expression of *Rp49*, and differences in transcript abundance were computed using the delta-delta-Ct method. The oligos used are listed in Table S2.

**Western blotting.** Sets of three adults were homogenized in 100 μl 2× SDS sample buffer (Bio-Rad #1610737) containing 5% (355 mM) 2-mercaptoethanol, heat-treated at 95° for 5 min, and centrifuged at maximum speed for 5 min to pellet debris. Fifteen microliters of each supernatant was loaded into a precast 4–20% polyacrylamide gradient gel (Bio-Rad). After electrophoresis, proteins were transferred to polyvinylidene difluoride membrane (PVDF, Millipore) using the Trans-Blot Turbo Transfer Pack (Bio-Rad #1704158) on the Trans-Blot Turbo apparatus (Bio-Rad). The membranes were cut between the Akt- and H3-adjacent mass markers to permit separate staining of these proteins without cross-talk or repeated stripping, because all antibodies were raised in rabbits. The uncropped blots are shown at the end of the Supplementary Information file. The membranes were blocked in blocking buffer (LI-COR, #927-40100), and the upper part was incubated with rabbit anti-phospho-Akt (Cell Signaling Technology, #4054) and the bottom, with rabbit anti-histone H3 (Abcam, #ab1791), both diluted 1:1000 in

blocking buffer + 0.2% Tween-20. The membranes were rinsed and incubated with IRDye IRDye 800CW anti-rabbit (LI-COR #925-32210) secondary, diluted 1:10,000; bands were visualized using the Odyssey Fc imaging system (LI-COR). The upper membrane portions were stripped, and total Akt was stained using a similar procedure with rabbit anti-Akt (Cell Signaling Technology, #4691, 1:1000).

**Feeding assays.** Short-term food intake was measured using a dye-feeding assay[84,85]. At the time of the main morning meal (after lights-on in the incubator, ZT0), flies were quickly transferred to nutrient-balanced sugar-yeast food[86] (90 g/L sucrose, 80 g/L yeast, 10 g/L agar, with 1 mL/L propionic acid and 1 g/L methyl-4-hydroxybenzoate) containing 0.5% erioglaucine (blue) dye (Sigma-Aldrich, #861146) and allowed to feed for 30 min. Other flies were allowed to feed on undyed medium for use in baselining spectrophotometric measurements. Ten sets of 3 animals for each genotype were homogenized in 50 µl of 50 mM PB, pH 7.5, using a TissueLyser LT (Qiagen) bead mill with 5-mm stainless-steel beads (Qiagen, #69989). Insoluble debris was pelleted by centrifugation, 50 µl of the supernatant was transferred to a 384-well plate well, and absorbance at 629 nm was measured using an Ensight multi-mode plate reader (PerkinElmer). Readings were calibrated against an erioglaucine standard curve to compute the amount of food consumed per fly per 30 min. Longer-term food consumption was monitored using the CAFE assay[46]. Each fly was placed into a 2-ml Eppendorf tube with a 5-µl microcapillary inserted through a hole in the lid; the capillary was filled with liquid sugar-yeast medium[46] (50 g/L sucrose, 50 g/L yeast extract, with 1 mL/L propionic acid and 1 g/L methyl-4-hydroxybenzoate). To minimize evaporation, capillary-equipped tubes were kept in a moist chamber, and fly-free tubes were used to control for the level of evaporation. The amount of food consumed was determined by measuring the food level within the capillary tube every 24 h for 3 days. Fifteen flies were individually scored for each genotype. To monitor the performance of feeding-related behaviors, rather than the consumed amounts, we used the FLIC assay[47]. *Drosophila* Feeding Monitors (Sable Systems, US) were kept in a 29-degree incubator at 70% humidity with a 12-h light cycle. Individual flies were placed in the monitors the night before the assay to allow them to acclimate; at the following subjective dawn, fresh sugar-water medium (10% sucrose) was added, and feeding behaviors were recorded for 24 h. Physical contacts between the flies and the sugar-water feeding medium were recorded using the manufacturer's software and analyzed by a package in the *R* analysis environment (https://github.com/PletcherLab/FLIC_R_Code)[47].

**Sleep assays.** Sleep assays were carried out using the *Drosophila* Activity Monitor system and the associated DAMSystem3 software (v3.10.7; TriKinetics, Waltham, MA). Individual male or female flies (8 days old: 3 days at 18 degrees post-eclosion to complete the transition to adult physiology, followed by 5 days of 29-degree pretreatment to induce RNAi) were transferred to 2-mm-inner-diameter glass tubes mounted in DAM2 monitors; one end of each tube was sealed using a 250-µL individual PCR tube filled with sugar food (5% sucrose and 1% agar in water), and the other end was plugged with plastic foam. Monitors were placed within a 29-degree incubator (60% humidity) with a 12-h light/dark cycle. The first day of data, reflecting behavioral acclimation to the environment, was discarded. Activity was recorded for 24 h from subjective dawn; at the second dawn, the food-containing PCR tubes were replaced with similar tubes containing only 1% agar, and the subsequent 24 h of behavior under starvation was recorded. Sleep was defined as five minutes of quiescence. Data were analyzed using pySolo[87] (v1.1) and custom scripts written in the MATLAB environment (The MathWorks, Natick, MA).

**Metabolite measurements.** Bulk tri-/-diacylglycerides and glycogen were measured using protocols described in detail previously[86,88]. For each sample, 4 adult flies were homogenized in PBS buffer with 0.05% Tween-20 (Sigma, #1379) in a TissueLyser LT (Qiagen) bead mill with 5-mm stainless-steel beads. An aliquot of homogenate was used for bicinchonic-acid protein-level determination using commercial components (Sigma, #B9643, #C2284, and #P0914). Glycogen content was assayed by hydrolyzing glycogen into glucose with amyloglucosidase (Sigma, #A7420) and measuring the liberated glucose using a colorimetric assay kit (Sigma, #GAGO20). For determination of circulating glucose, hemolymph was extracted[86] from adult females before and after starvation, and glucose was measured using the colorimetric assay (Sigma, #GAGO20). Tri- and diacylglycerides were measured by cleaving their ester bonds using triglyceride reagent (Sigma, #T2449) to liberate glycerol, which was measured using Free Glycerol Reagent (Sigma, #F6428) in a colorimetric assay. Absorbances from both assays were read at 540 nm using an Ensight multimode plate reader (PerkinElmer). TAG and glycogen were quantified using corresponding glucose and glycerol standard curves. All measurements were normalized to the number of animals used for each sample.

**Optogenetic activation of AstC cells.** Optogenetic experiments made use of the sensitive blue-light-activated channelrhodopsin ChR2^XXL [36]. Standard lab diet was re-liquified by microwaving, and all-*trans*-retinal (Sigma, #R2500) was added to 100 µM final concentration from a 50-mM ethanol stock. Ethanol alone was added to control medium. Animals were grown in darkness, and adult females (*AstC>ChR2^XXL* or *AstC>ChR2^XXL, AstC-RNAi*) were transferred to food containing trans-retinal 5–7 days after eclosion, one day before the experiment began.

Animals (10 each) were mounted in a dark room under dim red light in 35-mm glass-bottomed culture dishes (MatTek #P35G-0-10-C), in which the 10-mm glass window was covered with clear double-sided adhesive tape. Flies were further immobilized by covering the microwell with a glass cover slide fixed in place with tape. A Zeiss LSM 900 confocal microscope with Zen software acquired a tiled image of the entire window using 630-nm red light with a 5x objective, to create an overview image of the mounted flies without activating the expressed ChR2^XXL. Against this overview image, the tiling and time-series functions of the microscope were used to illuminate either each animal's head or abdomen, as appropriate, with intense, focused 475-nm blue LED light for 4 s every 40 s for 4 min, a regime that was repeated every 30 min for three hours. Following this treatment, the animals were unmounted and frozen for glycogen analysis as described above.

**Statistics.** All statistics were computed using the Prism software package (GraphPad). Starvation–survival curves were analyzed using Kaplan–Meier log-rank tests. Other types of data were assessed for normality using Anderson–Darling, D'Agostino and Pearson, Shapiro–Wilk, and Kolmogorov–Smirnov checks before statistical tests of significance were run. Multiple comparisons were analyzed using ANOVA and Kruskal–Wallis tests, as appropriate, and pairwise comparisons were made using two-tailed *t*-tests for normally distributed data or two-tailed Mann–Whitney *U* tests for other data. For experiments with multiple variables (e.g., genotype and diet), a two-way ANOVA was performed to assess interactions, and Bonferroni's multiple comparisons were conducted as a post hoc test for comparisons between groups. Significance is shown in figures as ns, $p > 0.05$; *$p \leq 0.05$; **$p \leq 0.01$; ***$p \leq 0.001$.

**Reporting summary.** Further information on research design is available in the Nature Research Reporting Summary linked to this article.

## Data availability

All data generated or analyzed during this study are included in this published article, its Supplementary information files, and in the Source Data files, except for raw behavioral data (Fig. 5 and Supplementary Fig. 9), which are available from the corresponding author on reasonable request. Source data are provided with this paper.

## Code availability

The custom MATLAB script used for sleep analysis in this study is publicly available at https://zenodo.org/record/5772445.YbNYAb3MKUk.

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

## Acknowledgements

Anti-AKH was a generous gift of Jae Park (University of Tennessee). Anti-DILP2 was a kind gift of Ernst Hafen (ETH Zurich). Anti-AstC and anti-DILP3 were kind gifts of Jan Veenstra (University of Bordeaux). Meet Zandawala kindly gave AstC as well. Shu Kondo, Pierre Léopold, Ryusuke Niwa, and Alessandro Scopelliti generously provided fly lines. We thank the University of Indiana Bloomington *Drosophila* Stock Center, the Vienna *Drosophila* Resource Center, and the Kyoto stock center for maintaining and providing fly lines, the University of Iowa Developmental Studies Hybridoma bank for producing and providing anti-Prospero, and AddGene for producing and providing pCFD6. This work was supported by Novo Nordisk Foundation grant NNF19OC0054632 and Lundbeck Foundation grant 2019-772 to K.R. T.K., M.T.N., and K.V.H. were supported by funding from the Villum Foundation (15365) and Danish Council for Independent Research Natural Sciences (9064-00009B) to K.V.H.

## Author contributions

M.J.T., S.N., and K.R. conceived and designed the study. O.K., T.K., N.A., L.N., A.M., S.N., K.V.H., M.J.T., and K.R. designed, performed, and analyzed experiments. M.T.N. performed experiments. M.L. analyzed data. M.J.T. and K.R. wrote the manuscript.

## Competing interests

The authors declare no competing interests.
