## [Peer Review File · Nature Communications]

Reviewers' Comments:

Reviewer #1:

Remarks to the Author:

The paper is well written paper with very good methodology and interesting results. The authors use multiple techniques, investigating both the gut and the brain.

It has been shown that male Akhr-null mutants also are starvation resistant, have higher triglyceride levels and, similarly, reduce food intake during starvation (10.1242/jeb.016451). In this report, any defects in locomotor behaviour or circadian rhythm were observed. This data raises two concerns:

1) In Figure 5D, the authors show that total night sleep duration in fed EEC>Cas9, AstCKO flies is similar to control flies. However, it seems in figure 5C that daytime sleep is increased for both fed and starved AstCKO flies, especially at Zeitgeber 0, an observation that was not discussed by the authors. Remarkably, this is the time on which the feeding behaviour assays were performed. Thus, decreased locomotor activity and increased daytime sleep could be factors contributing to the starvation resistance and decreased feeding in female flies. The authors should address this issue.

2) In Figures 2B and 2C the authors show a small, but consistent increase across experiments in TAG levels in EEC>AstC-RNAi fed flies. This difference is not present in AKH>Dcr-2 AstC-R2-RNAi or AKH>Dcr-2 Dcr-2-R2-RNAi lines while flies were fed, which might indicate that AstC absence induces TAG accumulation independent of APC activation and AKH signalling. The authors should discuss these inconsistencies.

Moreover, the author deleted AstC in the EECs, but they did not investigate the efficiency of the knockout in EEC>Cas9, AstCKO compared to control (EEC>Cas9). This can be performed measuring the transcripts of AstC or immunostaining as they did with EEC>AstC-RNAi. We thought the authors could have used another AstC RNAi line, but this is in part covered with the Crisper/Cas9 mutation.

One sentence in the result section which is vague and confusing. (page 7, line 7 & 8) {suggesting that the depletion of AstC levels in the EECs in response to starvation can be attributed to increased release and not to decreased production} of what? The authors should clarify this.

Reviewer #2:

Remarks to the Author:

This is an interesting manuscript dealing with the metabolic homeostasis function of AstC-producing midgut enteroendocrine cells (EEC) in the fruit fly *Drosophila melanogaster*. The authors tried to genetically manipulate EEC that expresses Allatostatin C (AstC) and found that the AstC-EEC plays an important role in maintaining metabolic homeostasis and stimulating feeding activity during nutrient-deprived conditions. An important finding is that AstC-EECs regulate glucagon-like adipokinetic hormone (AKH)-producing cells (APCs) in the endocrine corpora cardiaca (CC) to activate AKH expression and secretion. The activated AKH system eventually mobilizes glucagon and triglycerides and promote feeding activity. The data are mostly solid, presented clearly, and discussed reasonably. However, the authors should consider addressing the following concerns.

1. To manipulate AstC-EECs, this study used *voila-GAL4*, spatial expression of which is quite broad and not limited to the intestinal EEC (Ref #61). It seems active not only in the brain but also in many other neurons in the VNC and peripheral organs. Authors need to examine anti-AstC and anti-EGFP staining of the brain and the VNC from EEC-GFP females. Besides, the brain images of Fig 2E do not show all AstC-positive somas, probably due to small image size. Please provide high-resolution images of the brain and the VNC from EEC-AstCKO (Cas9), EEC-AstC-RNAi, and their controls.

2. Related to the above comment, I feel the authors need to provide more convincing evidence

supporting their claim that AstC from EEC is responsible for all the phenotypes they described.

3. The RNAi data are not fully controlled. The UAS-RNAi elements alone should be examined for these experiments

4. UAS-Dicer2 was included in some RNAi experiments, but not in others. This is particularly worrisome, as the authors provide no assessment of the effectiveness of their RNAi knockdown.

5. In general, AstC is an inhibitory peptide. Likewise, somatostatin also inhibits glucagon secretion from the pancreas (Ref #42). Thus, I feel uneasy with the claim that AstC/AstC-R2 activates AKH cells directly. Further, the current data cannot completely rule out the possibility that AstC-EECs regulate AKH activity indirectly through other cells (i.e., neurons in the CNS or PNS). I noted strong AstC-R2-GAL4 activity in (seemingly) neural processes that wrap APCs (Fig. 3A). Thus, it is also possible that AstC from EEC inhibits the activity of AstC-R2-GAL4 neurons that wrap APCs and supply inhibitory tones to APCs via AstC. In this scenario, AstC from EEC de-represses-and seemingly activates- APCs by inhibiting the AstC-R2-positive neurons. Thus, I strongly urge authors to provide better evidence backing their extraordinary claim. For example, AstC treatment on the isolated CC from females that express AstC-R2 in APCs but not in all neurons induces either Ca²⁺ elevation/AKH secretion or AKH expression.

6. There are many AstC-R2 neurons in the brain. Do they contribute to the regulation of metabolic homeostasis (glycogen and triglyceride levels) and feeding behaviors?

7. This study reported metabolic phenotypes specific to the female. Thus, authors should address possible interactions between reproductive and metabolic processes (i.e, by evaluating AstC-EEC functions in females lacking the functional ovary).

8. The AstC system negatively regulates juvenile hormone (JH), essential for oogenesis progression. If AstC secreted by EEC can access CC hormonally, it should also be able to influence JH-producing corpora allata. Thus, authors should also evaluate the possible involvement of the JH system.

9. Authors claim that the AstC system prevents hypoglycemia in food-deprived conditions. However, females lacking AstC showed a better starvation resistance than those with AstC. Why has the fruit fly evolved to maintain metabolic homeostasis in the sacrifice of the starvation resistance, which they might need to cope with the scarce food conditions that occur quite often in the natural environment? What would be a consequence(s) of dysregulated glycemia in the fruit fly, and why does the fruit fly need to avoid it?

Minor comments

1. Previous studies reported that starvation increased anti-DILP activity in IPCs (Enell et al, 2010; Zhan et al, 2016). However, Fig. S3C, D showed no significant increase. I noted relatively large degrees of variations in anti-DILP activities. Secretory activity of IPC is subjected to the circadian oscillation. Thus, authors need to control the time of the experiment precisely.

2. Figure 5C. Even in the fed condition, EEC-AstCKO flies seem to show more day-time sleep than the controls. Also, the lack of difference between KO and the control groups in the night-time sleep is likely due to the ceiling effect. Thus, EEC-AstCKO flies seem to sleep more regardless of feeding states.

3. It appears that the authors measured CaLexA activity in R3 domains and anti-AstC activity in R3 and R4. Authors also mentioned, "This suggests that at least a subset of AstC-expressing EECs are activated when food is absent, consistent with the peptide's being released during these conditions." Is there any variation among midgut domains in terms of secretory or Ca²⁺ activity of AstC-EECs?

4. Ref#36 is for a generation of UAS-TSC1/2. Please cite the proper reference(s) for the statement, "This manipulation mimics nutrient deprivation in these cells in an otherwise nutrition-replete animal."

5. According to a recent study (Zhang et al., 2021), AstC seems expressed in more cells than " in a small set of neurons in the adult brain." I suggest revising the statement, "Previous work has shown that AstC is also produced in a small set of neurons in the adult brain".

6. All figures. Authors used different types of statistical methods depending on the dataset, and so they need to indicate the statistical method used for each dataset.

7. Scale bar is missing in all image data

8. Figure 2E. Need to revise, "Below, a rotated and enlarged view of gut stain"

9. Units are missing in 'TAG levels', 'Glycogen levels', 'Hemolymph glucose', etc. It seems most of the data in this study is presented as normalized. Need to explain how normalization was done.

10. Figure 5A. The values for ul/fly/day is much smaller than those for ul/fly/30min. Despite the difference in the measuring method used for each dataset, the difference is simply too large to be real.

Reviewer #3:

Remarks to the Author:

The Gut hormone AstC regulates food intake and metabolic homeostasis under nutrient stress

In this paper, Kubrak, Jensen, Ahrentlöv et al study the role and regulation of the neuroendocrine peptide and enteroendocrine cell (EE) produce hormone Allatostatin C (AstC) in nutrient sensing and regulation of metabolic homeostasis and feeding behaviour by the gut.

Through a series of experiments, they determine that, nutrient availability influences AstC secretion from EEs, which then influences glucagon-like AKH production to signal to the brain and energy storage tissues to regulate lipid and glucose homeostasis and feeding behaviour. Lack of nutrients favour AstC secretion, in part via TOR inhibition, to allow mobilization of energy stores through AKH production/secretion and signaling to metabolic tissues. Knocking down AstC impairs starvation-induced energy mobilization and renders animals resistant to the detrimental effect of the lack of nutrients.

Overall, this paper delivers an interesting message and the characterization of a previously unreported functions of AstC. However, there are key controls and experiments that must be performed to sustain the main conclusions/claims of the paper.

Major comments:

1- The biggest issue of this paper is that it does not properly address the gut specific effects of AstC. The authors fall short in addressing this point, which is essential to support the conclusions of the paper: that gut AstC is the sole or main source of hormone regulating systemic effects/responses to nutrient availability. The sole use of voila-gal4 is problematic as this driver is widely expressed in the brain. As the authors pointed AstC is also expressed in the adult brain, which has also been implicated in nutrient sensing in *Drosophila* (e.g <https://doi.org/10.1016/j.neuron.2015.05.032>). Therefore, it is critical that the authors unambiguously address this. Comments and suggestions to address this point:

a- AstC expression: The data presented in Figure 2D, E is not sufficient to demonstrate the lack of effect of voila-gal4 on the brain peptide. Detection of RNA changes from a small subpopulation of neurons is difficult when considering the whole mass of the brain/head tissue, where AstC transcript is underrepresented. Can the authors detect AstC knockdown from whole heads when overexpressing AstC-RNAi using a driver that targets these cells? (e.g. AstC-gal4). Immunofluorescence images in E need to be replaced by high magnification images showing co-localization between AstC and voila-gal4 cells and appropriate quantification of the peptide. Recent papers (Zhang et al., 2021; Diaz et al., 2019) show that AstC is expressed in relevant brain circadian clusters. The authors should perform counterstaining with, for example TIM antibody, to look that the neuropeptide levels in conditions as shown in 2E. The images shown in E appear to reflect an increase in AstC in the AstC-RNAi expressing animals? Minor comment: There is a $P=0.07$ on the top of the bar in 2B.

b- AstC regulation: is brain AstC regulated by nutrients? This should be addressed in a quantitative manner.

c- AstC function: Does brain only knockdown of AstC (e.g. nSyb-gal4 driven RNAi) affect any of the phenotypes described here? (e.g. starvation survival, lipid and glucose metabolism, feeding). Complementarily, a more restrictive GAL4 driver should be used to target the gut peptide. Have the authors tried to combine AstC-gal4 with a neuronal gal80? In all cases, authors need to carefully analyse the expression of the brain peptide by high resolution imaging and co-localization with the driver being used.

2- General comment for all functional assays: The authors should use both Gal4 and UAS controls for their behavioral and survival experiments.

3- Figure 1I: This experiment needs to be done with a more restrictive driver. Minimally, the authors should inhibit TOR using AstC-gal4 and assess brain and gut peptide levels. Is TOR impairment sufficient to recapitulate AstC phenotypes? Also, in their model, the authors proposed

that nutrients regulate TOR in AstC producing EE cells. But this has not been demonstrated. Is this indeed the case? This should be doable using TOR signaling reporters combined with a marker of AstC EE cells.

4- The authors associate changes in the levels of AstC in EE cells induced by nutrients as change in hormone secretion (e.g. less hormone in the gut, more secreted). This has not been demonstrated. Can the authors use enzymatic or Western blot approaches to assess peptide hormone levels in the haemolymph?

5- Figure 1G top and bottom panels. I do not see the difference in the CaLexAGFP quantified in 1H. In Figure 1H, the authors plot data from individual cells from 9 guts. How has this data been normalised? This should be clearly explained.

6- The staining presented in Figure 4A are not very clear. There is significant signal in the right panels outside the GFP area. What does this correspond to? The presence of negative/positive controls would be important.

7- In Figure 5D the authors analyse total amount of sleep at night, but data in 5C it looks like day sleeping is also affected in the fed condition, authors should add this analysis. In the same panel they quantify the amount of sleep with the downregulation of AstC-R2 in APCs; authors need to show also the sleep graph for these genotypes as in C.

Minor comments:

In all figures, please change "P" of p value for lower case. Regarding the statistics: if there are two variables with two levels, e.g: feeding (Fed vs Starved) and genotype (Ctrl vs RNAi), the authors need to use a Two-way ANOVA, assess the interaction and then establish which comparisons are relevant. There is inconsistency on what groups are being compared in the different figures (e.g. fed versus starved in ctrl and RNAi in some cases and fed versus starved within the same genotypes in others) which makes the data interpretation confusing. It would be helpful if the authors can clarify the statistical analysis performed adding the value of the statistic with the corresponding degrees of freedom, and which post-hoc analysis did they use.

Reviewer #1 (Remarks to authors)

The paper is well written paper with very good methodology and interesting results. The authors use multiple techniques, investigating both the gut and the brain. It has been shown that male Akhr-null mutants also are starvation resistant, have higher triglyceride levels and, similarly, reduce food intake during starvation (10.1242/jeb.016451). In this report, any defects in locomotor behaviour or circadian rhythm were observed. This data raises two concerns:

Response) The reviewer finds that our manuscript is well-written and interesting. We thank the reviewer for this remark and for his or her very insightful comments. The reviewer makes a good point about the inter-connection between sleep and feeding, two mutually exclusive behaviors. In parallel with this *AstC* work, we have performed a very large multi-dimensional screen of more than 200 hundred secreted factors (unpublished) for EEC-derived gut hormones that affect feeding and sleep, and we can see in the resulting data the possible connection between these two behaviors. Thus, the reviewer's comment is quite apt, and we explain below how we have addressed this point as well as the reviewer's other comments.

Reviewer point 1) In Figure 5D, the authors show that total night sleep duration in fed *EEC>Cas9, AstCKO* flies is similar to control flies. However, it seems in figure 5C that daytime sleep is increased for both fed and starved *AstCKO* flies, especially at Zeitgeber 0, an observation that was not discussed by the authors. Remarkably, this is the time on which the feeding behaviour assays were performed. Thus, decreased locomotor activity and increased daytime sleep could be factors contributing to the starvation resistance and decreased feeding in female flies. The authors should address this issue.

Response) The reviewer points out that although the night-time sleep patterns of *EEC>Cas9, AstC^{KO}* flies are similar to those of controls (formerly Fig. 5d; now Fig. 5e of the revised manuscript), these animals seem to sleep more during the day even when they are fed (formerly Fig. 5c; now 5d). The reviewer asks whether this increased sleep fraction could potentially contribute to the decrease in food intake during daytime, simply through the mutually exclusive nature of these behaviors (Fig. 5a and b). To gain insight into whether the observed decreased food intake is a consequence of increased sleep, we binned daytime sleep and feeding episodes into 4-hour intervals. This analysis shows that, in the fed state, *EEC>Cas9, AstC^{KO}* flies exhibit strongly reduced feeding during the middle of the day (Zeitgeber time 4-8), even though they display an amount of sleep-like quiescence similar to controls' (Supplementary Fig. 9c and d). These new data argue that loss of *AstC* in the gut affects feeding, at least in part, through a mechanism that is independent of sleep regulation. We have now added these new data and a discussion of this issue in the revised manuscript, and we have cited appropriate literature on the functional inter-connection between feeding and sleep behaviors.

We also appreciate the reviewer's comment addressing the potential link between sleep amount and the ability of animals to resist starvation. We are very interested in how these are related and are currently exploring this. As mentioned above, we have recently conducted a multi-dimensional screen of EEC-secreted factors and can, with confidence, relay the observation that indeed there may be a link – albeit in the direction opposite from expectations. Animals which are susceptible to starvation tend to have higher sleep both during both day and night. If necessary, we will confidentially share this information with the reviewer. However, the main point of our report is that *AstC* acts via AKH signaling to regulate starvation resistance. We have performed many additional new experiments to support this conclusion, as described above. We have used pan-neuronal *nSyb-GAL4 (R57C10-GAL4)* to exclude phenotypic contributions from the nervous system – both from *AstC* released by neurons and from *AstC* signaling onto or within the nervous system (Supplementary Fig. 4a-e). Furthermore, adding pan-neuronal *GAL80 (R57C10-GAL80)* to *AstC-GAL4 (R57C10-G80, AstC>)* to inhibit *GAL4* activity in the nervous system further supports an EEC-specific origin of metabolically active *AstC* (Fig. 1g, 2h, and i). On this point, we have also made the important demonstration through *ex vivo* peptide application that *AstC* acts on the APCs to promote calcium activity and AKH release and that this effect is dependent on *AstC-R2* in the APCs (Fig. 4c). We have also performed tissue-transplantation experiments that indicate that the gut releases *AstC* in a nutrient-dependent manner *into the circulation*, through which it reaches

the APCs to promote AKH release (Supplementary Fig. 8c). These additional experiments in the revised manuscript strongly support our conclusion that the metabolic effects of AstC are mediated by AKH, which mainly affects starvation resistance through regulation of energy metabolism in the fat body. Since we show that AKH mediates effects of AstC on starvation resistance, we believe that this indicates that animals lacking AstC in EECs are resistant to starvation because AKH release is attenuated in these animals. We have also added a new optogenetic experiment that shows that exogenously induced AstC release from the gut rapidly triggers energy mobilization (Fig. 2j), which further argues that sleep is not the way by which *AstC* loss increases resistance to starvation. We believe it will be of interest for future studies to investigate the link between sleep, energy balance, and starvation resistance.

Reviewer point 2) In Figures 2B and 2C the authors show a small, but consistent increase across experiments in TAG levels in *EEC>AstC-RNAi* fed flies. This difference is not present in *AKH>Dcr-2 AstC-R2-RNAi* or *AKH>Dcr-2 Dcr-2-R2-RNAi* lines while flies were fed, which might indicate that AstC absence induces TAG accumulation independent of APC activation and AKH signalling. The authors should discuss these inconsistencies.

Response) The reviewer raises the point that TAG levels in *EEC>AstC-RNAi* animals are slightly increased in the fed state, a difference not observed in *AKH>Dcr-2, AstC-R2-RNAi*. We consistently see indication of a small increase in TAG levels in fed animals for some of our conditions. Even the one the reviewer mentions, *AKH>Dcr-2, AstC-R2-RNAi*, seems to have slightly, but not significantly, higher levels of TAG (Fig. 3c). However, the main effect of these manipulations appears during starvation, during which animals lacking *AstC* function in the EECs, similar to animals with loss of *AstC-R2* or *AKH* in the APCs, retain much higher levels of TAG after starvation, indicating that they mobilize and deplete their energy more slowly. We have now performed two-way ANOVA statistical analyses of these data according to the recommendation of reviewer #3. This analysis shows that animals lacking *AstC* in the EECs or *AstC-R2* or *AKH* in the APCs in all cases have significantly higher TAG and glycogen levels after starvation, with statistically significant interactions between diet and genotype that strongly support our conclusion that AstC signaling, via its effects on AKH secretion, is required during nutrient deprivation for energy mobilization. However, the tendency of animals to carry more TAG in the fed state is not significant with this highly stringent statistical analysis (Fig. 2b-d). We have also used an additional driver, *R57C10-G80, AstC>*, to specifically knock *AstC* down in the EECs and not in the CNS, and again with this driver we find much higher TAG levels after starvation in animals lacking *AstC* in the gut, while no significant difference was observed in the fed state (Fig. 2h). We use around 10 samples for metabolic measurements, but even with this number of replicates it is difficult to give statistical support to very small differences. Thus, although animals with deficient EEC AstC signaling may exhibit a minor elevation of TAG levels in the fed state, we cannot state that with rigor, and we believe it remains too speculative to discuss at length in the manuscript. We will of course include a discussion of this possibility should the reviewer prefer it.

Reviewer point 3) Moreover, the author deleted AstC in the EECs, but they did not investigate the efficiency of the knockout in *EEC>Cas9, AstCKO* compared to control (*EEC>Cas9*). This can be performed measuring the transcripts of AstC or immunostaining as they did with *EEC>AstC-RNAi*. We thought the authors could have used another AstC RNAi line, but this is in part covered with the Crispr/Cas9 mutation.

Response) We agree that it is a good idea to investigate the knockout efficiency of the *EEC>Cas9, AstC^{KO}*, and thus we have analyzed *AstC* transcript levels in midguts and CNS (brain and ventral nerve cord, VNC) of knockouts and controls (*EEC>Cas9*). These new data show that our somatic CRISPR knocks *AstC* out in the midgut without affecting *AstC* in the CNS (Supplementary Fig. 1k), similar to the *EEC>AstC-RNAi* manipulation (Fig. 2e). Furthermore, we have performed analysis of *AstC* expression to show that *AstC-GAL4>AstC-RNAi* can target *AstC* in both the CNS and the midgut (Supplementary Fig. 5a). RNAi driven by *R57C10-GAL4* (an *nSyb-GAL4* variant; the same enhancer fragment that drives this GAL4 is also used to express GAL80 in other genotypes to suppress neuronal GAL4) knocks *AstC* down in the CNS but, notably, not in the midgut (our data indicate that this driver might be a very useful for the community since other pan-neuronal drivers are also active in the EECs). Finally, we have also dissected the adipokinetic-hormone-producing cells (APCs) of the corpora cardiaca (CC) to show that *AKH-GAL4>AstC-R2-RNAi* and *AKH-GAL4>Cas9, AstC-R2^{KO}* efficiently reduce the expression of *AstC-R2* in these cells (Supplementary Fig. 6h). Thus, we have verified the knockdown and knockout efficiency for almost all conditions in the revised manuscript.

Reviewer point 4) One sentence in the result section which is vague and confusing. (page 7, line 7 & 8) (suggesting that the depletion of AstC levels in the EECs in response to starvation can be attributed to increased release and not to decreased production) of what? The authors should clarify this.

Response) We apologize for the confusing sentence. We have now clarified that this indicates AstC release.

Reviewer #2 (Remarks to authors)

This is an interesting manuscript dealing with the metabolic homeostasis function of AstC-producing midgut enteroendocrine cells (EEC) in the fruit fly *Drosophila melanogaster*. The authors tried to genetically manipulate EEC that expresses Allatostatin C (AstC) and found that the AstC-EEC plays an important role in maintaining metabolic homeostasis and stimulating feeding activity during nutrient-deprived conditions. An important finding is that AstC-EECs regulate glucagon-like adipokinetic hormone (AKH)-during nutrient-deprived conditions. An important finding is that AstC-EECs regulate glucagon-like adipokinetic hormone (AKH)-producing cells (APCs) in the endocrine corpora cardiaca (CC) to activate AKH expression and secretion. The activated AKH system eventually mobilizes glucagon and triglycerides and promote feeding activity. The data are mostly solid, presented clearly, and discussed reasonably. However, the authors should consider addressing the following concerns.

Response) We thank the reviewer for finding our manuscript interesting and for taking the time to provide many useful comments on our work. The reviewer finds our data mostly solid and clearly presented and discussed, and he or she raises some points that we have addressed as explained below. We have performed many additional experiments and included a substantial amount of new data in the revised manuscript to address these and other points [four new figures (Supplementary Fig. 2-5) and 46 new figure panels].

Reviewer points 1 and 2) 1. To manipulate AstC-EECs, this study used *voilà-GAL4*, spatial expression of which is quite broad and not limited to the intestinal EEC (Ref #61). It seems active not only in the brain but also in many other neurons in the VNC and peripheral organs. Authors need to examine anti-AstC and anti-EGFP staining of the brain and the VNC from EEC-GFP females. Besides, the brain images of Fig 2E do not show all AstC-positive somas, probably due to small image size. Please provide high-resolution images of the brain and the VNC from EEC-AstCKO (Cas9), EEC-AstC-RNAi, and their controls.

2. Related to the above comment, I feel the authors need to provide more convincing evidence supporting their claim that AstC from EEC is responsible for all the phenotypes they described.

Response) The reviewer points out that the *voilà-GAL4* (*voilà>*) driver (here referred to as *EEC>*) is not specific to the EECs but also drives some neuronal expression (point 1) and therefore asks for additional evidence to support the EEC-specificity of the observed AstC phenotypes (point 2). We apologize for not having been clear enough about showing that our conditions only affect gut AstC and not AstC in the brain or ventral nerve cord (VNC) in our original submission. We have now performed many additional experiments to show that AstC from the EECs, and not from the nervous system, is responsible for the observed phenotypes. We have stained brains and VNCs of *EEC>GFP* animals for AstC and GFP, as suggested by the reviewer. We have included high-resolution AiryScan2 images with zooms into specific regions in the nervous system, and dedicated two entirely new figures to provide evidence that there is no overlap between AstC and GFP in the brain or VNC of *EEC>GFP* animals (Supplementary Fig. 2 and 3). These results are very clear and show that although *voilà>* (*EEC>*) does exhibit some neuronal expression, it is not active in AstC-positive neurons and therefore should not affect AstC in the CNS.

In line with this finding, we now show by qPCR of dissected tissues that RNAi against *AstC* driven by *voilà>* (*EEC>*) only affects midgut *AstC* with no effect on transcript levels in the CNS (brain and VNC) (Fig. 2e). We thank the reviewer for pointing out that AstC is expressed not only in the brain, but also the VNC. Our analysis in the revised manuscript is therefore based on dissected brains and VNCs, not heads (containing only the brain) as in the original submission. As suggested by the reviewer, we have also performed this analysis for CRISPR-mediated knockout of *AstC*, and we have again found effects on *AstC* only in the midgut and not the CNS (Supplementary Fig. 1k). We have also now provided super-resolution images of brains, VNCs, and guts of *EEC>AstC-RNAi* animals obtained with a Zeiss LSM 900 confocal in “AiryScan 2” mode. These new high-resolution images show that knockdown of *AstC* with *voilà>* (*EEC>*), which eliminates AstC staining in the EECs, does not affect brain or VNC AstC levels (Fig. 2f), which are quantified in Supplementary Fig. 1l.

Together these quantifications of *AstC* expression in the CNS and midgut, combined with imaging of *AstC* immunostaining, provide strong evidence that *voilà*> (*EEC*>)-driven *AstC* manipulation only targets midgut *AstC* without affecting CNS levels. We appreciate the reviewer's comments and believe that these experiments provide the necessary evidence to address this point.

However, since we are aware that it is a general challenge in the field to find drivers that target EECs without effects on the CNS, owing to the cell-biological similarities between these cell types, we wished to provide additional lines of evidence to reinforce the EEC origin of *AstC*-mediated phenotypes, which also relates to and addresses the reviewer's point 2. To further demonstrate that gut-derived *AstC* is responsible for the observed phenotypes, we examined the metabolic effects of neuronal knockdown of *AstC*. First, we provide high-resolution AiryScan 2 images of midguts from animals in which *R57C10-GAL4* (*R57C10*>; the *Janelia* version of *nSyb-GAL4* enhanced for strong neuronal-specific activity) drives GFP expression. We find that this driver, unlike other common pan-neuronal drivers, *does not drive expression* in the *AstC*-positive EECs or indeed in any cells intrinsic to the gut, rather showing staining only in neuronal processes innervating the posterior midgut (Supplementary Fig. 4b). We show that pan-neuronal knockdown of *AstC* using this driver (*R57C10*>*AstC-RNAi*) strongly reduces *AstC* expression specifically in the CNS and, as expected, not in the midgut (Supplementary Fig. 4a). Next, we show that this strong neuronal knockdown of *AstC* is not responsible for the observed phenotypes, as this manipulation does not increase starvation resistance (Supplementary Fig. 4c) or mobilization of TAG (Supplementary Fig. 4) or glycogen (Supplementary Fig. 4e) during nutritional deprivation. These data provide strong evidence that neuronal *AstC* does not contribute in a major way to the observed metabolic phenotypes and thus indicate that gut-derived *AstC* is the source of these effects, which was the subject of the reviewer's second point.

To solidify this conclusion even further, we took advantage of the modular nature of the construction of the *R57C10*> driver and obtained animals in which the same *nSyb* enhancer elements are used to govern the expression of GAL80, a posttranslational inhibitor of GAL4 activity. Thus, *R57C10-GAL80* represses GAL4 activity throughout the nervous system without affecting expression elsewhere (such as the midgut). We found that *R57C10-GAL80, AstC*>*AstC-RNAi* animals – in which *AstC* is knocked down specifically in the *AstC*-positive EECs – exhibit increased resistance to starvation (Fig. 1g and Supplementary Fig. 1e), similar to the phenotype of *EEC*>*AstC-RNAi* knockdowns and *EEC*>*Cas9, AstC^{KO}* knockouts (Fig. 1b and c). The use of this independent driver (*R57C10-GAL80, AstC*>) strongly supports the EEC origin of *AstC*-mediated starvation resistance. We also examined the effect of this knockdown on energy mobilization and found that *R57C10-GAL80, AstC*>*AstC-RNAi* animals deplete their TAG and glycogen more slowly than controls (Fig. 2h and i), again phenocopying *voilà*> (*EEC*>) manipulations and showing that gut-derived *AstC* is responsible for the metabolic effects observed during nutrient deprivation.

Finally, to even further demonstrate the gut as the source of energy-mobilizing *AstC*, we employed an optogenetic approach. We expressed a light-activated Channelrhodopsin-2 (*ChR2^{XXL}*), which depolarizes cells upon light activation¹, to induce activation of either brain or gut *AstC* cells and thus to exogenously induce *AstC* release. We used *AstC*> to drive expression of this light-activated ion channel in *AstC*-positive cells of the CNS and the gut. We then used an automated microscope stage to deliver light pulses (4 seconds per animal every 40 seconds for 4 minutes, repeated every 30 minutes for three hours) precisely targeted at either the head or abdomen (where the midgut is located) of immobilized flies. In this way, we activated the *AstC*-expressing cells of either the brain or the midgut, and we measured the influence of this activation on stored glycogen, which is a fast-responding source of energy. Our results show that *AstC*>*ChR2^{XXL}* animals receiving abdominal illumination retained markedly lower glycogen compared to animals illuminated on the head (Fig. 2j), indicating that induction of *AstC* release from the gut promotes the depletion of glycogen. This effect was abrogated by simultaneous knockdown of *AstC*, showing that this effect is mediated by *AstC*. This indicated further that acute induction of gut *AstC* release causes mobilization of energy. Taken together, these additional data provide several lines of evidence strongly supporting our conclusion that EEC-derived *AstC* is responsible for all the observed phenotypes we discuss.

Reviewer point 3) The RNAi data are not fully controlled. The UAS-RNAi elements alone should be examined for these experiments

Response) The reviewer raises a point about the controls and asks to include the *UAS-RNAi* elements alone. We understand the reviewer's point and apologize for not having been clear enough about this in the first place. Let me first explain why our experiments are very well controlled and how we ensure that all the genetic

manipulations are performed following what we believe is the best standard. We have an in-house platform for all the genetics that we do, so that all our fly lines have similar “genetic background”. More precisely, we maintain a *w¹¹¹⁸*-derived genetic population that contains a small pool of genetic variation (*i.e.*, the stock is not completely isogenized). This allows us to control genetic background without potentially introducing homozygous artifacts through isogenization. Indeed, we have a dedicated genetics expert that makes all our transgenes to ensure that every element is maintained in this same genetic population background. All transgenic lines that were built and used for our study were made by this genetic expert. For all our transgenes, even when we combine transgenes on different chromosomes or when we recombine them on the same chromosome, we cross them into this same genetic background. In our revised manuscript, we have also included a new experiment to further show that *AstC* from EECs promotes food intake. In this experiment, we had to combine 7 different transgenes (*R57C10-GAL80*; *AstC::T2A::GAL4/UAS-TrpA1*, *UAS-GAL4*; *Tub-GAL80^{ts}*, *UAS-Cas9*, *UAS-AstC^{KO}*) in the same animal – *R57C10-GAL80* to suppress neuronal *GAL4* activity of *AstC::T2A::GAL4*; *UAS-TrpA1* to activate *AstC*-producing EECs using elevated temperature; *UAS-Cas9* and *UAS-AstC^{KO}* to knock out *AstC* in the EECs; and *UAS-GAL4* to maintain *GAL4* expression (to drive *UAS-TrpA1*) in the *AstC*-positive EECs even after knockout of the *AstC* locus, since that also disrupts the *AstC::T2A::GAL4* driver, a knock-in into the native *AstC* locus (Fig. 5c). This is genetically quite challenging and as far as we know may indeed be the highest number of independent transgenes combined into a single fly. Again, we did this by crosses that involved backcrossing into the same “genetic population background”. We make sure that animals even have similar X chromosomes by the combination of females versus males in these crosses. We also always remove, for example, *yellow* from these backgrounds, because it may have effects on physiological phenotypes such as metabolism and size regulation.

We understand that in some cases it can be good to test the *UAS-RNAi* elements alone, but if everything else is kept in the same background, the *UAS-RNAi* elements are not the best controls because they are *not* in the same genetic population background. The *GAL4* drivers are always the best controls when they are crossed to the genetic background in which the *UAS-RNAi* lines are kept, which we have done. Thus, the *GAL4* driver controls are genetically almost identical with the *RNAi* animals, except for the *RNAi* construct itself. The *GAL4* drivers are also better to control for any effects of the *GAL4* protein itself. In fact, we have in our study performed an *RNAi* screen including 35 *RNAi* lines targeting different secreted factors (Fig. 1a). These *RNAi* lines all have the same genetic background, since they are from the same KK collection (Vienna *Drosophila* *RNAi* collection, VDRC). Thus, when *AstC* knockdown produces a phenotype among all these lines, which are almost identical genetically except only for the actual *RNAi* construct, genetic background effects can be excluded. In that case, there is only one remaining concern and that is the possibility of off-target effects, in which the *RNAi* produces a phenotype due to unintended gene knockdown. To exclude this possibility, it is important to use a second independent *RNAi* or a somatic CRISPR knockout method (as also pointed out by reviewer #1). We have done this, showing with different *RNAi* lines and with CRISPR constructs that our effects are not due to off-targets. Thus we have excluded both genetic-background and off-target effects. As we mention above, we have also performed a very large screen of secreted factors from the EECs in adult flies covering more than 200 genes (unpublished). In this multidimensional screen, *AstC* knockdown produced the second-strongest increase in starvation resistance among all the tested genes (again effects were female-specific), indicating that *AstC* is absolutely one of the most important gut hormones during nutritional challenges and again also reducing the possibility of genetic background effects, since all the *RNAi* lines used in this large screen were of similar genetic background.

If transgenes are maintained with similar genetic backgrounds, as we have ensured in our work, the reason for using the *UAS-RNAi* elements alone is to ensure that the *UAS-RNAi* elements do not have some “leaky” expression based on chromosomal position effects even without the *GAL4* drivers. However, we show that different *RNAi* lines from different *RNAi* collections (GD, KK and/or TRiP lines) and CRISPR-mediated gRNA deletion (landed on different places) phenocopy one another, excluding this possibility. Again, it is also worth mentioning our two *RNAi* screens, discussed above, since all the *RNAi* constructs are landed in the same genomic site, excluding position effects. We also believe that our carefulness with genetics is apparent in our results, in which we usually see relatively small variations. In other words, we want to convey to the reviewer that we have been extremely careful to ensure that everything has been done under very-well-controlled conditions in all our experiments. We thank the reviewer for bringing up this point, and we have now clarified the genetics in the Methods section of the revised manuscript and explained how lines were backcrossed to introduce this similar genetic background. We think for the future that it might be very useful

for the field to agree on certain standards, and perhaps have some genetic guidelines for how to standardize genetic backgrounds. For example, some studies chose to completely isogenize flies, which we believe is not the best method, since these animals presumably become homozygous mutants for several genetic loci, which makes them “sick”. Indeed, a set of isogenized lines is used as a screen tool such as *Drosophila* Genetic Reference Panel (DGRP)². Instead, it is better to keep the genetic background very similar (“near-isogenic”), but still with a small amount of genetic variation to avoid this problem and still to standardize the genetic background, as we have done. We have started to prepare a manuscript discussing about genetic background controlling approaches and methods, which we hope will be well-received and useful by the community.

All this being said, we have actually also tested the *UAS-RNAi* elements alone as the reviewer suggested, to exclude any contributions of these in our experiments and to address the reviewer’s point in the most straightforward manner. We now show that these constructs do not contribute to the observed phenotypes (Supplementary Fig. 1b, e, h, 5b, c, 6d, j, 9a, and b). All these additional data support our conclusions.

Reviewer point 4) *UAS-Dicer2* was included in some RNAi experiments, but not in others. This is particularly worrisome, as the authors provide no assessment of the effectiveness of their RNAi knockdown.

Response) The reviewer’s point about the effectiveness of RNAi knockdown is well-received. We have now assessed knockdown efficiency for the experiment in which *UAS-Dicer-2* (*Dcr-2*) was included to enhance the RNAi effect. This was done on dissected APCs (CC) and shows that *AKH>Dcr-2, AstC-R2-RNAi* efficiently knocks *AstC-R2* down in the APCs (Supplementary Fig. 6h). Furthermore, we have also determined the CRISPR knockout efficiency of *AstC-R2* in this tissue and found that CRISPR-mediated *AstC-R2^{KO}* also reduces expression of *AstC-R2* in the APCs (Fig. S6H). In addition to this, we have analyzed both *AstC* knockdown and knockout in midguts and CNS (brain and ventral nerve cord, VNC) using the *EEC>* driver. These new data clearly show the efficiency of the *AstC* knockdown/knockout, and that these manipulations only affect midgut *AstC* without influencing transcript levels in the CNS (Fig. 2e and Supplementary Fig. 1k). Furthermore, we have also performed analysis to show knockdown efficiency with *AstC-GAL4* (Supplementary Fig. 5a). We also now show that pan-neuronal *R57C10-GAL4* (a variant of *nSyb-GAL4*, whose regulatory elements we have also used to drive GAL80 to suppress neuronal GAL4 activity) efficiently knocks *AstC* down in the CNS and not the midgut (Supplementary Fig. 4a). All these new data included in the revised manuscript show efficiency of knockdown/knockout and address this point raised by the reviewer.

Reviewer point 5) In general, *AstC* is an inhibitory peptide. Likewise, somatostatin also inhibits glucagon secretion from the pancreas (Ref #42). Thus, I feel uneasy with the claim that *AstC/AstC-R2* activates AKH cells directly. Further, the current data cannot completely rule out the possibility that *AstC-EECs* regulate AKH activity indirectly through other cells (i.e., neurons in the CNS or PNS). I noted strong *AstC-R2-GAL4* activity in (seemingly) neural processes that wrap APCs (Fig. 3A). Thus, it is also possible that *AstC* from EEC inhibits the activity of *AstC-R2-GAL4* neurons that wrap APCs and supply inhibitory tones to APCs via *AstC*. In this scenario, *AstC* from EEC de-represses-and seemingly activates- APCs by inhibiting the *AstC-R2*-positive neurons. Thus, I strongly urge authors to provide better evidence backing their extraordinary claim. For example, *AstC* treatment on the isolated CC from females that express *AstC-R2* in APCs but not in all neurons induces either Ca²⁺ elevation/AKH secretion or AKH expression.

Response) The reviewer mentions that *AstC* is known for its inhibitory actions, which is true. However, *AstC* acts through a G-protein-coupled receptor (GPCR) – *AstC-R2* in this case – and GPCRs can couple to different downstream G proteins (*G α s*, *G α q*, or *G α i/o*) that elicit stimulatory or inhibitory responses. Recently, evidence has accumulated that the same GPCR, in particular in the insulin-producing cells (IPCs) and adipokinetic-hormone-producing cells (APCs), couples to different G proteins and therefore has opposite effects on insulin and AKH. This presumably allows a single hormone to activate insulin secretion while downregulating AKH secretion after a meal or, in an opposite manner, to induce AKH release while inhibiting insulin secretion during fasting. Thus, from a physiological point of view, the ability of a peptide hormone to elicit opposing response on insulin and AKH appears quite efficient. Recently, a study published in *Nature* showed that short Neuropeptide F (sNPF) stimulates IPC activity and insulin secretion, while suppressing APC activity and AKH release³. This effect seems to be mediated through the coupling of the sNPF receptor, a GPCR, to the

stimulatory $G\alpha_q$ in the IPCs and the inhibitory $G\alpha_i/o$ in the APCs. Likewise, another recent study found that neuropeptide F (NPF, not related to sNPF) works in opposite directions on the IPCs and APCs, enhancing insulin secretion while suppressing AKH release⁴. We also note that a very recent study published this year finds that certain clock neurons produce AstC, which acts negatively on the IPCs to inhibit insulin release⁵. Thus, with our study showing that AstC *promotes* AKH release, this would be the third recent example of a peptide's having opposite effects on insulin and AKH. Our study provides many lines of evidence that AstC promotes AKH signaling: (a) phenotypes such as starvation resistance and metabolic balance indicate that *AstC* loss induces *AKH* loss-of-function phenotypes (Fig. 1a-c, g, 2a-d, h-j, 3b-d, g, and f); (b) genetic rescue experiments show that restoring *AKH* expression rescues phenotypes caused by loss of *AstC* signaling (Fig. 3g and h); and (c) *in-vivo* genetic experiments show that *AstC* loss in the EECs leads to retention of AKH (Fig. 4e) and that lack of AstC signaling in the APCs reduces calcium activity and leads to AKH retention (Fig. 4a, b, and d).

Beyond these data, the most direct experiment to demonstrate the valence of AstC signaling – direct application of peptide to isolated tissue – was suggested by the reviewer, and we have now performed this experiment. Application of synthetic AstC peptide to isolated female APCs (CC) in hemolymph-like buffer increases APC calcium activity and leads to a decrease in AKH levels, indicating increased AKH release (Fig. 4c). Moreover, knockdown of *AstC-R2* in the APCs using the *AKH*> driver completely abrogates these effects, demonstrating that AstC acts directly on the APCs (CC) in an AstC-R2-dependent manner to activate cellular calcium-signaling activity and AKH release. Furthermore, we have now provided evidence that even if the APCs are carefully removed from wild-type animals and transplanted into a different adult female host, starvation of the host leads to AKH release from the APCs, indicating the humoral nature of the inducing signal. This effect requires AstC expression in the host-animal EECs, since EEC-specific *AstC* knockout in the host animals abolishes this effect (Supplementary Fig. 8c). Thus, we have now performed the additional experiments suggested by the reviewer that provide strong additional evidence that AstC acts on the APCs and stimulates activity leading to AKH release.

Reviewer point 6) There are many AstC-R2 neurons in the brain. Do they contribute to the regulation of metabolic homeostasis (glycogen and triglyceride levels) and feeding behaviors?

Response) To address whether AstC-R2-expressing neurons in the brain contribute to the regulation of metabolic homeostasis or feeding behaviors, we used the pan-neuronal *R57C10-GAL4* (*R57C10*>) driver, which we show causes efficient knockdown in the nervous system without affecting midgut *AstC* expression (Supplementary Fig. 4a and b). Neuronal knockdown of *AstC* or *AstC-R2* did not increase resistance to starvation, and these animals depleted their energy stores (TAG and glycogen) during starvation at similar rates to controls (Supplementary Fig. 4-e). We also found that neuronal loss of *AstC* or *AstC-R2* did not change food intake (Supplementary Fig. 9a). These data together indicate that AstC- and AstC-R2-expressing neurons do not contribute in a major way to the regulation of metabolic homeostasis and feeding behaviors in our experiments, addressing the point raised by the reviewer.

Reviewer point 7) This study reported metabolic phenotypes specific to the female. Thus, authors should address possible interactions between reproductive and metabolic processes (i.e, by evaluating AstC-EEC functions in females lacking the functional ovary).

Response) The reviewer suggests addressing the possible interaction between reproductive and metabolic processes in females. To do address this point, we investigated whether EEC-derived AstC affects female fecundity. We found that knockdown of *AstC* in the EECs did not influence female egg laying (Supplementary Fig. 5d), indicating that gut-derived AstC does not affect fecundity. We have now included these new data along with a discussion of the results in the revised manuscript.

Reviewer point 8) The AstC system negatively regulates juvenile hormone (JH), essential for oogenesis progression. If AstC secreted by EEC can access CC hormonally, it should also be able to influence JH-producing corpora allata. Thus, authors should also evaluate the possible involvement of the JH system.

Response) AstC and several other peptides (AstA and AstB) were originally characterized for their ability to regulate juvenile hormone (JH) release from the corpora allata (CA) in non-*Drosophila* insects. The main role of JH during development is to prevent metamorphosis until the final larval instar; however, for *Drosophila*, unlike many other insects, JH is not important for this developmental transition⁶. Many of these peptide hormones, originally characterized as regulators of JH, has since been shown to have other functions in *Drosophila*. Of course, that does not suggest that they cannot also regulate JH in flies. Although we feel that it is a little bit beyond the scope of this work, since we focus on the AstC-AKH connection, we have addressed the point raised by the reviewer by knocking down expression of the two AstC receptors, AstC-R1 and AstC-R2, in the CA. Adult-restricted receptor knockdown driven by the CA-specific *Aug21-GAL4*⁷ with *Tub-GAL80^{ts}* did not increase starvation resistance, indicating that AstC-mediated modulation of JH signaling does not contribute to the observed AstC-knockdown phenotypes. These new data (Supplementary Fig. 6f) are included and discussed in the revised manuscript.

Reviewer point 9) Authors claim that the AstC system prevents hypoglycemia in food-deprived conditions. However, females lacking AstC showed a better starvation resistance than those with AstC. Why has the fruit fly evolved to maintain metabolic homeostasis in the sacrifice of the starvation resistance, which they might need to cope with the scarce food conditions that occur quite often in the natural environment? What would be a consequence(s) of dysregulated glycemia in the fruit fly, and why does the fruit fly need to avoid it?

Response) The reviewer raised the question of why flies evolved to maintain metabolic homeostasis through the sacrifice of starvation resistance, since the AstC system prevents hypoglycemia under food-deprivation conditions. It is well known (and we have also shown it here) that flies lacking AKH or its receptor AkhR become highly resistant to starvation because they mobilize their energy, and thus deplete their stores, more slowly⁸⁻¹⁰. Although this may seem like an increase in fitness from an evolutionary standpoint, this observation applies of course under laboratory conditions. In normal ecological settings, when an animal encounters local nutritional scarcity, it is possible that there are other food sources nearby, so the starving animal must initiate its food-seeking behaviors, including flying from one place to another, and therefore it also needs to mobilize energy to support its searching behaviors. Both of these effects are promoted by AstC and AKH. However, an animal in a starvation experiment will never find new a food source no matter how hard it searches – it is stoppered into a vial, after all – and will fruitlessly burn all of its energy and die sooner. Under these conditions, it becomes disadvantageous to activate AstC or AKH signaling, while in the wild, these systems must be beneficial to survival since they were selected for and were not lost during evolution (both AstC and AKH are conserved). Similar phenotypes are also observed for other pathways that are important for fat mobilization – e.g., loss of the lipase Brummer⁸ also leads to increased starvation resistance in the laboratory.

Minor Comments:

Reviewer minor point 1) Previous studies reported that starvation increased anti-DILP activity in IPCs (Enell et al, 2010; Zhan et al, 2016). However, Fig. S3C, D showed no significant increase. I noted relatively large degrees of variations in anti-DILP activities. Secretory activity of IPC is subjected to the circadian oscillation. Thus, authors need to control the time of the experiment precisely.

Response) The reviewer points out that previous studies have found increased DILP stain in the IPCs after starvation. While this is true, most of these studies have focused on starvation-induced DILP retention during development, and thus this effect has been most strongly shown for larvae. Fewer studies have investigated effects of starvation on anti-DILP activity in adults, and these have looked at longer-term effects (usually 24 h) and mostly in males¹¹⁻¹³. Our work is shorter-term (6 h) and involves females, whose metabolism is quite different from males'. In two studies mentioned by the reviewer (Enell et al. 2010 and Zhan et al. 2016), both looked at longer-term starvation (12-24 h) on DILP2, compared to our analysis, and one only used males. Thus, a likely explanation is that females may not respond by retaining substantial amounts of insulin quickly enough for us to see a change after 6 hours' food deprivation. Our Western blot of pAKT, a downstream target of insulin signaling⁶, supports the idea that 6 hours' starvation does not change systemic insulin signaling (Supplementary Fig. 7e), suggesting that it does not reduce circulating DILPs. We note that another study of females also indicates that dietary restriction does not change DILP2 in adult females¹⁴. We performed all the

stains on tissues dissected at the same time of the day (in fact also the same day), so the experiments are controlled for the time of day.

Reviewer minor point 2) Figure 5C. Even in the fed condition, EEC-AstCKO flies seem to show more day-time sleep than the controls. Also, the lack of difference between KO and the control groups in the night-time sleep is likely due to the ceiling effect. Thus, EEC-AstCKO flies seem to sleep more regardless of feeding states.

Response) We have addressed the increased daytime sleep of *AstC^{KO}* flies in our response to Reviewer #1's first point, which we kindly invite the reviewer to read, and we have now discussed this in our revised manuscript.

Reviewer minor point 3) It appears that the authors measured CaLexA activity in R3 domains and anti-AstC activity in R3 and R4. Authors also mentioned, "This suggests that at least a subset of AstC-expressing EECs are activated when food is absent, consistent with the peptide's being released during these conditions." Is there any variation among midgut domains in terms of secretory or Ca²⁺ activity of AstC-EECs?

Response) The reviewer asks whether there is any variation among midgut domains in terms of calcium or AstC secretory activity. This is an interesting question! However, we have focused on the middle-midgut region R3 because we (and previous investigators) find that most AstC immunoreactivity falls within this region, suggesting that it is a major domain for AstC-positive EECs. We do not know for sure whether there is variation in other midgut domains, but it will be interesting for future studies to investigate.

Reviewer minor point 4) Ref#36 is for a generation of UAS-TSC1/2. Please cite the proper reference(s) for the statement, "This manipulation mimics nutrient deprivation in these cells in an otherwise nutrition-replete animal."

Response) We thank the reviewer for pointing out that a proper reference is required for the statement. We have now modified the statement and added an appropriate reference.

Reviewer minor point 5) According to a recent study (Zhang et al., 2021), AstC seems expressed in more cells than "in a small set of neurons in the adult brain." I suggest revising the statement, "Previous work has shown that AstC is also produced in a small set of neurons in the adult brain".

Response) We agree with the reviewer and have now revised the statement to say that AstC is produced in a "subset" of neurons, instead of a "small set".

Reviewer minor point 6) All figures. Authors used different types of statistical methods depending on the dataset, and so they need to indicate the statistical method used for each dataset.

Response) We have now clearly described the different types of statistical analysis performed for each dataset in the figure legends.

Reviewer minor point 7) Scale bar is missing in all image data

Response) We thank the reviewer for bringing this to our attention. Scale bars are now added to all images.

Reviewer minor point 8) Figure 2E. Need to revise, "Below, a rotated and enlarged view of gut stain"

Response) Figure 2E: We have revised the sentence, and the view is no longer rotated.

Reviewer minor points 9 and 10) 9. Units are missing in 'TAG levels', 'Glycogen levels', 'Hemolymph glucose', etc. It seems most of the data in this study is presented as normalized. Need to explain how normalization was done.

10. Figure 5A. The values for ul/fly/day is much smaller than those for ul/fly/30min. Despite the difference in the measuring method used for each dataset, the difference is simply too large to be real.

Response) Again, we thank the reviewer for bringing this our attention. We have now specified in every figure that TAG and glycogen levels are represented as relative units, and the revised manuscript describes how measurements were normalized. We have corrected “ul/fly/30 min” to “nl/fly/30 min”, which explains why the volumes consumed appeared unreasonably large.

Reviewer #3 (Remarks to authors)

In this paper, Kubrak, Jensen, Ahrentlöv et al study the role and regulation of the neuroendocrine peptide and enteroendocrine cell (EE) produce hormone Allatostatin C (AstC) in nutrient sensing and regulation of metabolic homeostasis and feeding behaviour by the gut. Through a series of experiments, they determine that, nutrient availability influences AstC secretion from EEs, which then influences glucagon-like AKH production to signal to the brain and energy storage tissues to regulate lipid and glucose homeostasis and feeding behaviour. Lack of nutrients favour AstC secretion, in part via TOR inhibition, to allow mobilization of homeostasis and feeding behaviour. Lack of nutrients favour AstC secretion, in part via TOR inhibition, to allow mobilization of energy stores through AKH production/secretion and signaling to metabolic tissues. Knocking down AstC impairs starvation-induced energy mobilization and renders animals resistant to the detrimental effect of the lack of nutrients. Overall, this paper delivers an interesting message and the characterization of a previously unreported functions of AstC. However, there are key controls and experiments that must be performed to sustain the main conclusions/claims of the paper.

Response) The reviewer finds that our manuscript delivers an interesting story, and we thank the reviewer for that view as well as for taking the time to rigorously comment on our work, which has been very useful. Below we describe the many additional experiments and revisions we have performed to address all the reviewer’s points:

Major comments:

Reviewer point 1, A) The biggest issue of this paper is that it does not properly address the gut specific effects of AstC. The authors fall short in addressing this point, which is essential to support the conclusions of the paper: that gut AstC is the sole or main source of hormone regulating systemic effects/responses to nutrient availability. The sole use of *voilà-gal4* is problematic as this driver is widely expressed in the brain. As the authors pointed AstC is also expressed in the adult brain, which has also been implicated in nutrient sensing in *Drosophila* (e.g <https://doi.org/10.1016/j.neuron.2015.05.032>). Therefore, it is critical that the authors unambiguously address this. Comments and suggestions to address this point:

A) AstC expression: The data presented in Figure 2D, E is not sufficient to demonstrate the lack of effect of *voilà-gal4* on the brain peptide. Detection of RNA changes from a small subpopulation of neurons is difficult when considering the whole mass of the brain/head tissue, where AstC transcript is underrepresented. Can the authors detect AstC knockdown from whole heads when overexpressing AstC-RNAi using a driver that targets these cells? (e.g. AstC-gal4). Immunofluorescence images in E need to be replaced by high magnification images showing co-localization between AstC and *voilà-gal4* cells and appropriate quantification of the peptide. Recent papers (Zhang et al., 2021; Diaz et al., 2019) show that AstC is expressed in relevant brain circadian clusters. The authors should perform counterstaining with, for example TIM antibody, to look that the neuropeptide levels in conditions as shown in 2E. The images shown in E appear to reflect an increase in AstC in the AstC-RNAi expressing animals? Minor comment: There is a $P=0.07$ on the top of the bar in 2B.

Response) The reviewer points out that *voilà-GAL4* driver (*voilà>*; which we here refer to as *EEC>*) is not specific to the enteroendocrine cells (EEC) and in fact is also active in neurons. The reviewer suggests providing additional evidence to support the conclusion that EEC-derived AstC alone regulates systemic effects and responses to nutrient availability. This point is well-received and we apologize for not having provided enough evidence in our original submission to show that our conditions only affect gut AstC. We explain below the many additional experiments that we have performed to demonstrate that EEC-derived AstC is responsible for all the observed effects to address this point:

A: The reviewer asks whether we can detect *AstC* knockdown in whole heads when using *AstC-GAL4*. Instead of looking at whole heads, which contain only the brain, we have dissected central nervous systems (brains and ventral nerve cords, VNCs), since a couple of neurons in the VNC also express *AstC*. Indeed, we see very strong knockdown of *AstC* expression (~ 95% reduction of transcripts) in the CNS using *AstC-GAL4*, which also targets midgut *AstC* with similar efficiency (Supplementary Fig. 5a). These new data relating to the reviewer's point are included in the revised manuscript and show efficiency of the RNAi as well as our ability to detect reduced expression in the CNS. Furthermore, we have provided data on CNS (brain and VNC) instead of whole heads that show that the *voilà*> (*EEC*>) driver with *AstC-RNAi* only affects midgut *AstC* levels, with no effects observed in the CNS (Fig. 2e). Furthermore, we have now also performed this analysis for CRISPR-mediated knockout of *AstC* with the same *voilà*> (*EEC*>) driver, which reaches the same conclusion: this targets only midgut and not CNS *AstC* (Supplementary Fig. 1k).

The reviewer also requests high-magnification images showing (the lack of) co-localization between *AstC* and *voilà*> (*EEC*>). We have now provided super-resolution images of brains, VNCs, and EECs obtained with a Zeiss LSM 900 confocal in "AiryScan 2" mode. We show high-res images of the CNS with zooms into specific regions, and we have dedicated two entirely new figures to showing that there is no overlap between *AstC* and GFP from *voilà*>*GFP* in the brain or VNC (Supplementary Fig. 2 and 3), as suggested by the reviewer. The results are very clear and show that although the *voilà*> (*EEC*>) driver is active in some neurons, it does not express GAL4 in *AstC*-positive neurons, and therefore *voilà*>-driven *AstC* manipulations should not affect *AstC* in the CNS. On the other hand, we now also show co-localization between *voilà*>*GFP* and *AstC* staining in the gut (Fig. 2G), demonstrating that the driver targets *AstC*-positive gut EECs, consistent with its ability to drive knockdown or knockout of *AstC* in the midgut (Fig. 2e and Supplementary Fig. 1k). Furthermore, we now provide magnified high-resolution AiryScan-2 images replacing those in the previous Fig. 2e (now Fig. 2f) that the reviewer mentions. These images, quantified in Supplementary Fig. 11, show that knockdown of *AstC* with *voilà*> (*EEC*>) does not affect brain or VNC *AstC* levels, as suggested by the reviewer. Together all these new images of brains, VNCs, and midguts provide strong evidence that *AstC*-specific manipulations driven by *voilà*> (*EEC*>) only target midgut *AstC* without affecting CNS levels, again consistent with our data showing that knockdown or knockout of *AstC* with this driver does not influence *AstC* transcript levels in the CNS (Fig. 2e and Supplementary Fig. 1k). We appreciate the reviewer's point and believe that these experiments provide the necessary evidence to address it. Minor comments: the "P=0.07" in the figure has been replaced with non-significant (ns), and we thank the reviewer for pointing this out.

Reviewer point 1, B) b- *AstC* regulation: is brain *AstC* regulated by nutrients? This should be addressed in a quantitative manner.

Response) The reviewer asks whether brain *AstC* is regulated by nutrients. We have now addressed this in a quantitative manner using qPCR on dissected tissues, which indicates that brain *AstC* transcript levels are not regulated by nutrient deprivation (Supplementary Fig. 1d), and we discuss this in our revised manuscript.

Reviewer point 1, C) c- *AstC* function: Does brain only knockdown of *AstC* (e.g. *nSyb-gal4* driven RNAi) affect any of the phenotypes described here? (e.g. starvation survival, lipid and glucose metabolism, feeding). Complementarily, a more restrictive GAL4 driver should be used to target the gut peptide. Have the authors tried to combine *AstC-gal4* with a neuronal *gal80*? In all cases, authors need to carefully analyse the expression of the brain peptide by high resolution imaging and co-localization with the driver being used.

Response) The reviewer asks whether brain-only knockdown of *AstC* (e.g., *nSyb-GAL4*-driven RNAi) contributes to any of the observed phenotypes. To further demonstrate that gut-derived *AstC* alone is responsible for the effects, we assessed the effects of neuronal *AstC* knockdown using the pan-neuronal *R57C10-GAL4* (*R57C10*>; this is the "Janelia" version of *nSyb-GAL4* enhanced for neuronal-specific activity) to knock *AstC* down in the nervous system, as suggested by the reviewer. First, we showed this strongly reduces *AstC* expression (~ 95% reduction of transcripts) specifically in the CNS (brain and VNC) without affecting the midgut (Supplementary Fig. 4a). Next, to show that the *R57C10*> driver does not express in the *AstC*-positive EECs or any other cells in the gut, we have also included high-resolution confocal AiryScan images with zoom-ins of midguts from *R57C10*>*GFP* animals stained for *AstC* and GFP (Supplementary Fig. 4b;

note that *R57C10*> drives *GFP* in neuronal processes innervating the posterior midgut, but not in the gut itself). To investigate whether brain AstC is responsible for any of the observed effects of nutrient availability, we then used *R57C10*>*AstC-RNAi* to specifically knock *AstC* down in the nervous system. We found that neuronal knockdown of *AstC* does not increase starvation resistance (Supplementary Fig. 4c) or mobilization of TAG (Supplementary Fig. 4d) or glycogen (Supplementary Fig. 4e) during nutritional deprivation. Together this provides strong evidence that neuronal AstC does not contribute to the phenotypes and metabolic effects we investigate, indicating that gut-derived AstC is the source of these effects.

To solidify this conclusion even further we used the complementary approach suggested by the reviewer with a more restrictive GAL4 driver in combination with neuronal GAL80. We took advantage of the neuronal-specific *R57C10*> enhancer element to express GAL80 (G80), a GAL4 inhibitor, in the nervous system and thereby limit the activity of the *AstC-GAL4* (*AstC*>) driver to the EECs. *R57C10-G80, AstC*>*AstC-RNAi* animals with *AstC* knockdown specifically in the AstC-producing EECs display increased resistance to starvation (Fig. 1g and Supplementary Fig. 1e), phenocopying *voilà*-driven *AstC* knockdown (*EEC*>*AstC-RNAi*) or knockout (*EEC*>*Cas9, AstC^{KO}*) (Fig. 1b and c). These new results strongly support the gut-specific origin of AstC in regard to increased starvation resistance. To further support this case, we examined the mobilization of energy using *R57C10-G80, AstC*> to express RNAi against *AstC* and found that animals with EEC-specific *AstC* knockdown deplete their TAG and glycogen stores more slowly than controls (Fig. 2h and i), again supporting the results obtained with the *voilà*> (*EEC*>) driver and reinforcing the conclusion that gut-derived AstC is responsible for the metabolic effects we have observed.

Finally, to demonstrate further the gut as the source of energy-mobilizing AstC, we employed an optogenetic approach. We expressed a light-activated Channelrhodopsin-2 (*ChR2^{XXL}*), which depolarizes cells upon light activation¹, to induce activation of either brain or gut AstC cells and thus to exogenously induce AstC release. We used *AstC*> to drive expression of this light-activated ion channel in AstC-positive cells of the CNS and the gut. We then used an automated microscope stage to deliver light pulses (4 seconds per animal every 40 seconds for 4 minutes, repeated every 30 minutes for three hours) precisely targeted at either the head or abdomen (where the midgut is located) of immobilized flies. In this way, we activated the AstC-expressing cells of either the brain or the midgut, and we measured the influence of this activation on stored glycogen, which is a fast-responding source of energy. Our results show that *AstC*>*ChR2^{XXL}* animals receiving abdominal illumination retained markedly lower glycogen compared to animals illuminated on the head (Fig. 2j), indicating that induction of AstC release from the gut promotes the depletion of glycogen. This effect was abrogated by simultaneous knockdown of *AstC*, showing that this effect is mediated by AstC. This indicated further that acute induction of gut AstC release causes mobilization of energy. Taken together, these additional data provide several lines of evidence strongly supporting our conclusion that EEC-derived AstC is responsible for all the observed phenotypes we discuss.

Reviewer point 2) General comment for all functional assays: The authors should use both Gal4 and UAS controls for their behavioral and survival experiments.

Response) The reviewer raises the points that both GAL4- and UAS-alone controls should be used for behavioral and survival experiments. We understand this point, and it is well-received. We apologize for not having been clear enough about how we carefully have controlled the genetics of our experiments. We would therefore first like to take the opportunity explain in detail how we ensured that all genetics were done according to what we believe is the best standard, and secondly how we have now included the UAS controls that the reviewer suggest (the same response as to reviewer's #2 point 3).

We have an in-house platform for all the genetics that we do, so that all our fly lines have similar "genetic background". More precisely, we maintain a *w¹¹¹⁸*-derived genetic population that contains a small pool of genetic variation (*i.e.*, the stock is not completely isogenized). This allows us to control genetic background without potentially introducing homozygous artifacts through isogenization. Indeed, we have a dedicated genetics expert that makes all our transgenes to ensure that every element is maintained in this same genetic population background. All transgenic lines that were built and used for our study were made by this genetic

expert. For all our transgenes, even when we combine transgenes on different chromosomes or when we recombine them on the same chromosome, we cross them into this same genetic background. In our revised manuscript, we have also included a new experiment to further show that *AstC* from EECs promotes food intake. In this experiment, we had to combine 7 different transgenes (*R57C10-GAL80*; *AstC::T2A::GAL4/UAS-TrpA1*, *UAS-GAL4*; *Tub-GAL80^{ts}*, *UAS-Cas9*, *UAS-AstC^{KO}*) in the same animal – *R57C10-GAL80* to suppress neuronal *GAL4* activity of *AstC::T2A::GAL4*; *UAS-TrpA1* to activate *AstC*-producing EECs using elevated temperature; *UAS-Cas9* and *UAS-AstC^{KO}* to knock out *AstC* in the EECs; and *UAS-GAL4* to maintain *GAL4* expression (to drive *UAS-TrpA1*) in the *AstC*-positive EECs even after knockout of the *AstC* locus, since that also disrupts the *AstC::T2A::GAL4* driver, a knock-in into the native *AstC* locus (Fig. 5C). This is genetically quite challenging and as far as we know may indeed be the highest number of independent transgenes combined into a single fly. Again, we did this by crosses that involved backcrossing into the same “genetic population background”. We make sure that animals even have similar X chromosomes by the combination of females versus males in these crosses. We also always remove, for example, *yellow* from these backgrounds, because it may have effects on physiological phenotypes such as metabolism and size regulation.

We understand that in some cases it can be good to test the *UAS-RNAi* elements alone, but if everything else is kept in the same background, the *UAS-RNAi* elements are not the best controls because they are *not* in the same genetic population background. The *GAL4* drivers are always the best controls when they are crossed to the genetic background in which the *UAS-RNAi* lines are kept, which we have done. Thus, the *GAL4* driver controls are genetically almost identical with the *RNAi* animals, except for the *RNAi* construct itself. The *GAL4* drivers are also better to control for any effects of the *GAL4* protein itself. In fact, we have in our study performed an *RNAi* screen including 35 *RNAi* lines targeting different secreted factors (Fig. 1A). These *RNAi* lines all have the same genetic background, since they are from the same KK collection (Vienna *Drosophila* *RNAi* collection, VDRC). Thus, when *AstC* knockdown produces a phenotype among all these lines, which are almost identical genetically except only for the actual *RNAi* construct, genetic background effects can be excluded. In that case, there is only one remaining concern and that is the possibility of off-target effects, in which the *RNAi* produces a phenotype due to unintended gene knockdown. To exclude this possibility, it is important to use a second independent *RNAi* or a somatic CRISPR knockout method (as also pointed out by reviewer #1). We have done this, showing with different *RNAi* lines and with CRISPR constructs that our effects are not due to off-targets. Thus we have excluded both genetic-background and off-target effects. As we mention above, we have also performed a very large screen of secreted factors from the EECs in adult flies covering more than 200 genes (unpublished). In this multidimensional screen, *AstC* knockdown produced the second-strongest increase in starvation resistance among all the tested genes (again effects were female-specific), indicating that *AstC* is absolutely one of the most important gut hormones during nutritional challenges and again also reducing the possibility of genetic background effects, since all the *RNAi* lines used in this large screen were of similar genetic background.

If transgenes are maintained with similar genetic backgrounds, as we have ensured in our work, the reason for using the *UAS-RNAi* elements alone is to ensure that the *UAS-RNAi* elements do not have some “leaky” expression based on chromosomal position effects even without the *GAL4* drivers. However, we show that different *RNAi* lines from different *RNAi* collections (GD, KK and/or TRiP lines) and CRISPR-mediated gRNA deletion (landed on different places) phenocopy one another, excluding this possibility. Again, it is also worth mentioning our two *RNAi* screens, discussed above, since all the *RNAi* constructs are landed in the same genomic site, excluding position effects. We also believe that our carefulness with genetics is apparent in our results, in which we usually see relatively small variations. In other words, we want to convey to the reviewer that we have been extremely careful to ensure that everything has been done under very-well-controlled conditions in all our experiments. We thank the reviewer for bringing up this point, and we have now clarified the genetics in the Methods section of the revised manuscript and explained how lines were backcrossed to introduce this similar genetic background. We think for the future that it might be very useful for the field to agree on certain standards, and perhaps have some genetic guidelines for how to standardize genetic backgrounds. For example, some studies chose to completely isogenize flies, which we believe is not the best method, since these animals presumably becomes homozygous mutants for several genetic loci, which

makes them “sick”. Indeed, a set of isogenized lines is used as a screen tool such as *Drosophila* Genetic Reference Panel (DGRP)². Instead, it is better to keep the genetic background very similar (“near-isogenic”), but still with a small amount of genetic variation to avoid this problem and still to standardize the genetic background, as we have done. We have started to prepare a manuscript discussing about genetic background controlling approaches and methods, which we hope will be well-received and useful by the community.

All this being said, we have actually also tested the *UAS-RNAi* elements alone as the reviewer suggested, to exclude any contributions of these in our experiments and to address the reviewer’s point in the most straightforward manner. We now show that these constructs do not contribute to the observed phenotypes (Supplementary Fig. 1b, e, h, 5b, c, 6d, j, 9a, and b). All these additional data support our conclusions.

Reviewer point 3) Figure 1I: This experiment needs to be done with a more restrictive driver. Minimally, the authors should inhibit TOR using *AstC-gal4* and assess brain and gut peptide levels. Is TOR impairment sufficient to recapitulate *AstC* phenotypes? Also, in their model, the authors proposed that nutrients regulate TOR in *AstC* producing EE cells. But this has not been demonstrated. Is this indeed the case? This should be doable using TOR signaling reporters combined with a marker of *AstC* EE cells.

Response) The reviewer requests that the experiment in Figure 1I be done with a more-restrictive driver such as *AstC-GAL4* to inhibit TOR activity only in the cells of interest. We have now performed the experiment as suggested by the reviewer; we used *AstC-GAL4* in combination with *Tub-GAL80^{ts}* to express *TSC1/2* and induce inhibition of TOR only in *AstC*-positive cells. This experiment with the more restricted *GAL4* driver produced results similar to those obtained with the previous broader manipulation, showing that inhibition of TOR leads to strong reduction in *AstC* immunoreactivity in the EECs. These new data are included and discussed in the revised manuscript (Fig. 1I and k) to address the reviewer’s point. The reviewer also suggests using TOR reporters, but the problem is that these are very limited. We work a lot with TOR, and we can honestly say that it is not easy to find broadly useful reporters. While some reagents work for Western blots (e.g., anti-phospho-S6-kinase), it is much more difficult when it comes to ones that work for immunohistochemistry (IHC) with EECs, which are needed to perform the experiment suggested by the reviewer. We developed an antibody to detect phosphorylated S6, a downstream target of TOR and S6K, as a reporter of TOR, which has previously been used successfully for IHC in some situations¹⁵. We confirmed that this antibody can work for some tissues such as fat body and prothoracic gland, but the gut is more difficult to stain so we have not included these data. However, we believe that our *AstC-GAL4* experiment with inhibition of TOR addresses the reviewer’s point.

Reviewer point 4) The authors associate changes in the levels of *AstC* in EE cells induced by nutrients as change in hormone secretion (e.g. less hormone in the gut, more secreted). This has not been demonstrated. Can the authors use enzymatic or Western blot approaches to assess peptide hormone levels in the haemolymph?

Response) The reviewer asks whether enzymatic or Western blots can be used to assess *AstC* peptide hormone levels in the hemolymph to definitively link nutrient-induced changes in the levels of *AstC* in the EECs with secretion. We have performed immunostaining to show retention or release of *AstC* and combined that with transcriptional analysis to show that *AstC* transcripts are not reduced during nutrient deprivation, when *AstC* peptide levels decline, indicating that the decline in staining is not due to decreased expression, but can be attributed to increased release. Since it is almost impossible to detect small peptides like *AstC* in the hemolymph, our approach is the standard one used in the field. The level of these peptides is very low, and hemolymph volumes are tiny, making standard ELISA and most Western-blot methods very difficult or infeasible. Through our colleagues we are aware of one other attempt to measure a small peptide, neuropeptide F (NPF), with an ELISA-based method in *Drosophila*. In that study, the author uses a more sensitive ELISA method called DELFIA (also called time-resolved fluoroimmunoassay, TRF or TRFIA). However, after following up it seemed likely that what was detected with this method was not specific, so they unfortunately had to remove it before the publication of their work¹⁶. It is also worth mentioning that NPF is a larger peptide than *AstC*. We have tried hard to find ways of detecting peptides in minute amounts of hemolymph. Together with the largest pharmaceutical company in the authors’ country of Denmark (Novo Nordisk A/S), which is

the world's largest producer of insulin (a peptide hormone), we have tried to develop an "AlphaLISA"-based method, potentially offering much greater sensitivity than standard ELISA and even the DELPHIA methods employed for NPF. Unfortunately, despite our efforts we have so far not been successful in making this assay work. We are also aware that others are trying to solve this problem. The Perrimon lab is generating a suite of transgenic animals in which peptides carry different tags, but this approach also seems to be problematic. We have also tried mass-spectrometric proteomics approaches together with a leading expert, but thus far this technique has not permitted detection of these small peptides.

We therefore devised an alternative and technically quite challenging approach to show that AstC release is induced by nutritional changes and that this peptide acts systemically via the circulation (*i.e.*, the hemolymph). In our work, we find that AstC promotes adipokinetic hormone (AKH) release from AKH-producing cells (APCs) of the CC. We rationalized that if APCs (CC) are transplanted into the thorax (the normal anatomical location) of a second adult "host" fly, these cells can only be reached by circulating AstC (flies have an open circulatory system in which the hemolymph occupies the body cavity where it surrounds the organs). Thus, effects observed in the implanted tissue would indicate whether AstC is released from EECs of the host animal into the hemolymph and thus address one part of the reviewer's point. To address the other part – whether AstC release is nutrient-dependent – we used as host females that had been either fed or starved, with or without EEC-specific *AstC* knockout, to demonstrate that these effects are mediated by AstC. We then recovered these transplanted APCs and measured AKH levels. Our results show that APCs transplanted into starved host animals retain lower AKH levels, indicating increased AKH release (Supplementary Fig. 8c). This effect is abrogated by knockdown of *AstC* in the host EECs, demonstrating that it is induced by AstC. We also now performed *ex vivo* application of synthetic AstC peptide to dissected APCs and observed the same effects on these cells as we observe *in vivo* after starvation in an EEC AstC-dependent manner – induction of AKH release (Fig. 4c and d). Together these experiments suggest that AstC is released into the hemolymph in a nutrient-dependent manner and acts as a humoral factor on the APCs to promote AKH release, which we believe addresses the reviewer's point.

Reviewer point 5) Figure 1G top and bottom panels. I do not see the difference in the CaLexAGFP quantified in 1H. In Figure 1H, the authors plot data from individual cells from 9 guts. How has this data been normalised? This should be clearly explained.

Response) The reviewer points out that visual differences between the images in (the former) Figure 1g are not obvious, in light of the quantified differences between them (these are now Fig. 1h and 1i). We have therefore replaced the images with more-representative ones and clearly described how data were normalized and quantified in the Methods section, to address this point.

Reviewer point 6) The staining presented in Figure 4A are not very clear. There is significant signal in the right panels outside the GFP area. What does this correspond to? The presence of negative/positive controls would be important.

Response) The reviewer points out the staining in Figure 4a is not very clear. This is a ligand-binding assay, in which labelled ligand binds to unfixed living tissue expressing the cognate receptor. The signal is therefore not very clear, and we apologize for the technical limitations. We believe that the reduction in binding to the APCs when the receptor is knocked down serves as a control, showing that the binding is dependent on the receptor. The reviewer also refers to signal outside the GFP area, and we do not know the exact nature of this signal, but we can say for sure that the AstC signal binding to the APCs is AstC-R2-dependent. However, we have shown through multiple other approaches that AstC acts directly on the APCs through its receptor AstC-R2, so this ligand-binding assay is somewhat supplementary, and we have therefore moved it from the main figure to the Supplementary fig. 8a. If required, we can completely omit it.

Reviewer point 7) In Figure 5D the authors analyse total amount of sleep at night, but data in 5C it looks like day sleeping is also affected in the fed condition, authors should add this analysis. In the same panel they quantify the amount of sleep with the downregulation of AstC-R2 in APCs; authors need to show also the sleep graph for these genotypes as in C.

Response) We have now included the analysis of total amount of daytime sleep and the sleep graphs for *AstC-R2* knockdown in the APCs as suggested by the reviewer (Supplementary Fig. 9b and e) in the revised manuscript.

Minor Comments:

In all figures, please change “P” of p value for lower case. Regarding the statistics: if there are two variables with two levels, e.g: feeding (Fed vs Starved) and genotype (Ctrl vs RNAi), the authors need to use a Two-way ANOVA, assess the interaction and then establish which comparisons are relevant. There is inconsistency on what groups are being compared in the different figures (e.g. fed versus starved in ctrl and RNAi in some cases and fed versus starved within the same genotypes in others) which makes the data interpretation confusing. It would be helpful if the authors can clarify the statistical analysis performed adding the value of the statistic with the corresponding degrees of freedom, and which post-hoc analysis did they use.

Response) We have now changed all “P” to lower case “p” in the figures. Regarding statistics, we have now used two-way ANOVA to perform statistical analysis and assess interactions when analyzing experiments with two variables, such as diet versus genotype, as suggested by the reviewer. The interactions and specific statistics, including post hoc test, used in each figure are now specified in that figure’s legend.

In summary, we have added a substantial amount of data to address all the comments made by the reviewers. We believe that this has significantly improved our work, and we are thankful for reviewers’ time and their insightful comments. Importantly, we a provided several further lines of evidence strengthening the proposed mechanism that EEC-derived AstC coordinates food intake and mobilization of energy through interorgan signaling, acting via AstC-R2 in the APCs to regulate AKH release. We hope that our thoroughly revised manuscript is now suitable for publication, and we are pleased to submit it for your kind consideration.

As always, please do not hesitate to contact me if you need further information.

Sincerely yours,

Kim Rewitz

References

- 1 Dawydow, A. *et al.* Channelrhodopsin-2-XXL, a powerful optogenetic tool for low-light applications. *Proc Natl Acad Sci U S A* **111**, 13972-13977, doi:10.1073/pnas.1408269111 (2014).
- 2 Mackay, T. F. *et al.* The *Drosophila melanogaster* Genetic Reference Panel. *Nature* **482**, 173-178, doi:10.1038/nature10811 (2012).
- 3 Oh, Y. *et al.* A glucose-sensing neuron pair regulates insulin and glucagon in *Drosophila*. *Nature* **574**, 559-564, doi:10.1038/s41586-019-1675-4 (2019).
- 4 Yoshinari, Y. *et al.* The sugar-responsive enteroendocrine neuropeptide F regulates lipid metabolism through glucagon-like and insulin-like hormones in *Drosophila melanogaster*. *bioRxiv*, 2021.2005.2017.444252, doi:10.1101/2021.05.17.444252 (2021).
- 5 Zhang, C. *et al.* The neuropeptide allatostatin C from clock-associated DN1p neurons generates the circadian rhythm for oogenesis. *Proc Natl Acad Sci U S A* **118**, doi:10.1073/pnas.2016878118 (2021).
- 6 Texada, M. J., Koyama, T. & Rewitz, K. Regulation of Body Size and Growth Control. *Genetics* **216**, 269-313, doi:10.1534/genetics.120.303095 (2020).
- 7 Mirth, C., Truman, J. W. & Riddiford, L. M. The role of the prothoracic gland in determining critical weight for metamorphosis in *Drosophila melanogaster*. *Curr Biol* **15**, 1796-1807, doi:10.1016/j.cub.2005.09.017 (2005).
- 8 Gronke, S. *et al.* Dual lipolytic control of body fat storage and mobilization in *Drosophila*. *PLoS Biol* **5**, e137, doi:10.1371/journal.pbio.0050137 (2007).

- 9 Koyama, T., Texada, M. J., Halberg, K. A. & Rewitz, K. Metabolism and growth adaptation to environmental conditions in *Drosophila*. *Cell Mol Life Sci* **77**, 4523-4551, doi:10.1007/s00018-020-03547-2 (2020).
- 10 Lee, G. & Park, J. H. Hemolymph sugar homeostasis and starvation-induced hyperactivity affected by genetic manipulations of the adipokinetic hormone-encoding gene in *Drosophila melanogaster*. *Genetics* **167**, 311-323, doi:10.1534/genetics.167.1.311 (2004).
- 11 Enell, L. E., Kapan, N., Soderberg, J. A., Kahsai, L. & Nassel, D. R. Insulin signaling, lifespan and stress resistance are modulated by metabotropic GABA receptors on insulin producing cells in the brain of *Drosophila*. *PLoS One* **5**, e15780, doi:10.1371/journal.pone.0015780 (2010).
- 12 Luo, J., Becnel, J., Nichols, C. D. & Nassel, D. R. Insulin-producing cells in the brain of adult *Drosophila* are regulated by the serotonin 5-HT1A receptor. *Cell Mol Life Sci* **69**, 471-484, doi:10.1007/s00018-011-0789-0 (2012).
- 13 Rajan, A. & Perrimon, N. *Drosophila* cytokine unpaired 2 regulates physiological homeostasis by remotely controlling insulin secretion. *Cell* **151**, 123-137, doi:10.1016/j.cell.2012.08.019 (2012).
- 14 Broughton, S. J. *et al.* DILP-producing median neurosecretory cells in the *Drosophila* brain mediate the response of lifespan to nutrition. *Aging Cell* **9**, 336-346, doi:10.1111/j.1474-9726.2010.00558.x (2010).
- 15 Romero-Pozuelo, J., Demetriades, C., Schroeder, P. & Teleman, A. A. CycD/Cdk4 and Discontinuities in Dpp Signaling Activate TORC1 in the *Drosophila* Wing Disc. *Dev Cell* **42**, 376-387 e375, doi:10.1016/j.devcel.2017.07.019 (2017).
- 16 Ameku, T. *et al.* Midgut-derived neuropeptide F controls germline stem cell proliferation in a mating-dependent manner. *PLoS Biol* **16**, e2005004, doi:10.1371/journal.pbio.2005004 (2018).

Reviewers' Comments:

Reviewer #1:

Remarks to the Author:

I am happy with the revisions.

Reviewer #2:

Remarks to the Author:

The revised manuscript addressed all issues raised in the initial review to satisfaction. I only have a few minor comments on the revision.

1. The authors clearly demonstrated that AstC from voila-Gal4-positive cells in the midgut is responsible for the phenotypes of interest. Having shown that voila-Gal4 has more broad CNS expression than AstC, however, it seems a bit misleading to refer voila-Gal4 as EEC-Gal4. This is potentially problematic for Fig 1a because EECs are not necessarily responsible for phenotypes of other neuropeptide-RNAi driven by voila-Gal4.

2. About newly added UAS-RNAi control data figure (S1b, S1e, S1h, S5b, S5c, S6d). It seems to me that data points of the test group in the supplemental figures compared with the UAS-RNAi alone control group are the same as those in the main figures (in the first submission) compared with Gal4 control. Did authors collect Gal4 alone, UAS alone, and test group (Gal4/UAS) data simultaneously, but present them separately in response to reviewers' comments? Otherwise, the authors need to state that UAS alone and test group data are collected in different cohorts of experiments in the corresponding figure legend.

-Typos

line 376: 3d and d -> 3b and d

Fig S5b, missing the statistical significance label in the starved group

Fig S6h, AstC expression -> AstC-R2 expression in the APCs

Reviewer #3:

Remarks to the Author:

The gut hormone Allatostatin C/Somatostatin regulates food intake and metabolic homeostasis under nutrient stress

This revised manuscript satisfactorily addresses most of my major comments. There are a couple that still remain outstanding and, which I deemed of sufficient importance to be addressed.

Major comments

1- In my Comment 1b made on the original manuscript, I asked whether AstC from the brain (please note that this refer to the protein as per the used nomenclature) was regulated by nutrients. Consistently, I asked the authors to assess this in a quantitative manner. In response to my request, the authors assessed mRNA levels of AstC in the CNS. The outcome of the brain peptide mRNA is not surprising as the authors show and emphasized in their paper that is it AstC protein and not mRNA what is regulated by nutrients, in the gut. Consistently, and to adequately address my question, they should provide immunofluorescence and quantitative data on the levels of AstC peptide in the brain of fed versus starved animals.

2- Similarly, in my Comment 3 made on the original manuscript I asked that, minimally, the authors should inhibit TOR using AstC-gal4 and assess brain and gut AstC protein levels. The levels of the brain peptide in this condition were not addressed in the revised manuscript. This is important as nutrients are known to regulate brain AstC. Furthermore, I requested that the authors use a TOR signaling reporter to assess signaling activation in AstC EE cells in response to nutrients. This is essential to substantiate one of the main conclusions and proposed model of the paper, namely, that nutrients regulate TOR in AstC producing EE cell and that influences AstC

secretion. The authors mention that there is a lack of reliable TOR signaling reporters, precluding them from addressing this question. I understand the limitations. However, in this case, the authors should re-consider/revise this part of their model as there is no sufficient evidence to support it. They should consider their hypothesis as only a possibility rather than a fact and account for alternatives. For example, TOR could be acting as a permissive signaling in the system.

Minor comments:

1- Material and methods are clearer now. The authors have carried proper statistical analysis with adequate tests. However, it would be useful to have at least as part of the source data file (if this is included in the submission) statistical information such as the value of the F-statistic for ANOVA (or U for Mann Whitney, if it is a nonparametric) and degrees of freedom and exact p values if different from the standard *: $p < 0.05$; **: $p < 0.01$; ***: $p < 0.001$.

2- In the legend for Figure 2, there is still "P" instead of p for p-values. They have changed the rest.

Please find below a point-by-point response to the issues raised by each reviewer.

Reviewer #1 (Reviewer's remark to authors)

I am happy with the revisions.

Response: We thank the reviewer for being happy with the revision.

Reviewer #2 (Reviewer's remark to authors)

The revised manuscript addressed all issues raised in the initial review to satisfaction. I only have a few minor comments on the revision.

Response: We thank the reviewer for finding that the revised manuscript addressed all the issues raised. We have addressed the referee's minor comments.

Point #1) The authors clearly demonstrated that AstC from *voila-Gal4*-positive cells in the midgut is responsible for the phenotypes of interest. Having shown that *voila-Gal4* has more broad CNS expression than AstC, however, it seems a bit misleading to refer *voila-Gal4* as EEC-Gal4. This is potentially problematic for Fig 1a because EECs are not necessarily responsible for phenotypes of other neuropeptide-RNAi driven by *voila-Gal4*.

Response: The reviewer suggests referring to *EEC-GAL4* as *voilà-GAL4*, since this driver is not restricted to enteroendocrine cells (EECs). We agree with this point and have changed *EEC-GAL4* to *voilà-GAL4* throughout the text and in the figures to address this point.

Point #2) About newly added UAS-RNAi control data figure (S1b, S1e, S1h, S5b, S5c, S6d). It seems to me that data points of the test group in the supplemental figures compared with the UAS-RNAi alone control group are the same as those in the main figures (in the first submission) compared with Gal4 control. Did authors collect Gal4 alone, UAS alone, and test group (Gal4/UAS) data simultaneously, but present them separately in response to reviewers' comments? Otherwise, the authors need to state that UAS alone and test group data are collected in different cohorts of experiments in the corresponding figure legend.

Response: The reviewer also asked us to state in the supplementary figures (S1b, S1e, S1h, S5b, S5c, and S6d) if data were collected in different cohorts. We have now stated whether this is the case.

-Typos

line 376: 3d and d -> 3b and d

Fig S5b, missing the statistical significance label in the starved group

Fig S6h, AstC expression -> AstC-R2 expression in the APCs

Response: We have fixed the typos and the missing figure label. We thank the reviewer for catching these.

Reviewer #3 (Reviewer's remark to authors)

The gut hormone Allatostatin C/Somatostatin regulates food intake and metabolic homeostasis under nutrient stress. This revised manuscript satisfactorily addresses most of my major comments. There are a couple that still remain outstanding and, which I deemed of sufficient importance to be addressed.

Response: The reviewer finds that our revised manuscript satisfactorily address most of the major comments, except two that relate to the question of whether nutrient signaling affects AstC from the brain, which we address in our response below.

Point #1) In my Comment 1b made on the original manuscript, I asked whether AstC from the brain (please note that this refer to the protein as per the used nomenclature) was regulated by nutrients. Consistently, I asked the authors to assess this in a quantitative manner. In response to my request, the authors assessed mRNA levels of AstC in the CNS. The outcome of the brain peptide mRNA is not surprising as the authors show and emphasized in their paper that is it AstC protein and not mRNA what is regulated by nutrients, in the gut. Consistently, and to adequately address my question, they should provide immunofluorescence and quantitative data on the levels of AstC peptide in the brain of fed versus starved animals.

Response: The reviewer points out that in addition to the assessment of *AstC* mRNA levels in the CNS, we should provide quantitative immunofluorescence data on AstC peptide levels in the CNS. We have now performed the experiments suggested by the reviewer by immunostaining and quantifying AstC peptide levels in the CNSes in fed and starved animals. We did not detect any difference between fed and starved conditions. This indicates that at least the 6-hour nutrient deprivation that activates AstC release for the EECs does not affect brain AstC peptide levels. These new data are included in Supplementary Fig. 1d and e. We have also consulted with our colleague Professor Dick Nässel at the University of Stockholm, who is a leading expert in *Drosophila* neuropeptides. Neither he nor we are aware of any studies directly showing that brain AstC is regulated by nutrients. We acknowledge that perhaps a longer starvation duration (the brain may react more slowly than the gut to nutritional changes) or different nutritional conditions may affect AstC dynamics in the brain. We have included a discussion about this in our revised manuscript that also makes this point, that it is possible that AstC in the brain is influenced by different nutritional conditions and that it will be interesting for future studies to further examine how the brain senses nutrition and whether it involves AstC.

Point #2) Similarly, in my Comment 3 made on the original manuscript I asked that, minimally, the authors should inhibit TOR using AstC-gal4 and assess brain and gut AstC protein levels. The levels of the brain peptide in this condition were not addressed in the revised manuscript. This is important as nutrients are known to regulate brain AstC. Furthermore, I requested that the authors use a TOR signaling reporter to assess signaling activation in AstC EE cells in response to nutrients. This is essential to substantiate one of the main conclusions and proposed model of the paper, namely, that nutrients regulate TOR in AstC producing EE cell and that influences AstC secretion. The authors mention that there is a lack or reliable TOR signaling reporters, precluding them from addressing this question. I understand the limitations. However, in this case, the authors should re-consider/revise this part of their model as there is no sufficient evidence to support it. They should

consider their hypothesis as only a possibility rather than a fact and account for alternatives. For example, TOR could be acting as a permissive signaling in the system.

Response: The reviewer's second point also relates to nutrient signaling and brain AstC protein levels. The reviewer asks for testing whether inhibition of TOR in AstC-producing neurons using the *AstC-GAL4* driver affects brain AstC protein levels. We have now performed this experiment by expressing *Tsc1+Tsc2* using *AstC-GAL4* and assessed brain AstC protein levels as suggested by the reviewer. As with the starvation experiment of point #1, we do not observe that the inhibition of TOR in *AstC*-expressing neurons affects AstC peptide levels in the CNS. This indicates that at least the TOR-inhibition manipulation that is associated with AstC release from the EECs does not influence AstC release from the brain. We have included these new data in Supplementary Fig. 1i and j, along with a short discussion in the text. The reviewer also suggests that we revise our manuscript and model to say that TOR inhibition could be acting as a permissive signal for AstC secretion from the EECs. We think this is a good point. Instead of proposing that nutrients regulate TOR in AstC-producing EECs and that this influences AstC release, we now propose that TOR is somehow involved in the regulation of AstC secretion and could be acting as a permissive signal along with other factors as suggested by the reviewer. We have revised the text, including the abstract, and the model accordingly to address this point.

Minor comments:

1. Material and methods are clearer now. The authors have carried proper statistical analysis with adequate tests. However, it would be useful to have at least as part of the source data file (if this is included in the submission) statistical information such as the value of the F-statistic for ANOVA (or U for Mann Whitney, if it is a nonparametric) and degrees of freedom and exact p values if different from the standard *: $p < 0.05$; **: $p < 0.01$; ***: $p < 0.001$.

Response: We have provided a source data file with all the raw data and stated "n numbers" in the legends of each figure, along with the specific statistical analysis used for each graph as suggested by the reviewer.

2. In the legend for Figure 2, there is still "P" instead of p for p-values. They have changed the rest.

Response: We thank the reviewer for catching this. We have now changed "P" to "p" in the legend of figure 2.

In summary, we have addressed all the comments made by the reviewers and hope that our revised manuscript is now suitable for publication, and we are pleased to submit it for your kind consideration – hoping for an editorial decision, if possible.

As always, please do not hesitate to contact me if you need further information.

Sincerely yours,

Kim Rewitz

Reviewers' Comments:

Reviewer #2:

Remarks to the Author:

All of my concerns were addressed satisfactorily.

Reviewer #3:

Remarks to the Author:

The revised manuscript now fully addressed all my original comments.